# CoCoEdit: Content-Consistent Image Editing via Region Regularized Reinforcement Learning

Yuhui Wu [1 2]  Chenxi Xie [1 2]  Ruibin Li [1]  Liyi Chen [1]  Qiaosi Yi [1 2]  Lei Zhang [1 2]

## Abstract

Image editing has achieved impressive results with the development of large-scale generative models. However, existing models mainly focus on the editing effects of intended objects and regions, often leading to unwanted changes in unintended regions. We present a post-training framework for **Co**ntent-**Co**nsistent **Edit**ing (**CoCoEdit**) via region regularized reinforcement learning. We first augment existing editing datasets with refined instructions and masks, from which 40K diverse and high quality samples are curated as training set. We then introduce a pixel-level similarity reward to complement MLLM-based rewards, enabling models to ensure both editing quality and content consistency during the editing process. To overcome the spatial-agnostic nature of the rewards, we propose a region-based regularizer, aiming to preserve non-edited regions for high-reward samples while encouraging editing effects for low-reward samples. For evaluation, we annotate editing masks for GEdit-Bench and ImgEdit-Bench, introducing pixel-level similarity metrics to measure content consistency and editing quality. Applying CoCoEdit to Qwen-Image-Edit and FLUX.1 Kontext, we achieve not only competitive editing scores with state-of-the-art models, but also significantly better content consistency, measured by PSNR/SSIM metrics and human subjective ratings. Codes, data and models of CoCoEdit can be found at CoCoEdit.

## 1. Introduction

Leveraging large-scale training data and potent generative backbones, current large editing models, including both unified generation-editing models (Tian et al., 2025; Wei

[1]The Hong Kong Polytechnic University, Hong Kong; [2]OPPO Research Institute, ShenZhen, China. Correspondence to: Lei Zhang <cslzhang@comp.polyu.edu.hk>.

*Proceedings of the $43^{rd}$ International Conference on Machine Learning*, Seoul, South Korea. PMLR 306, 2026. Copyright 2026 by the author(s).

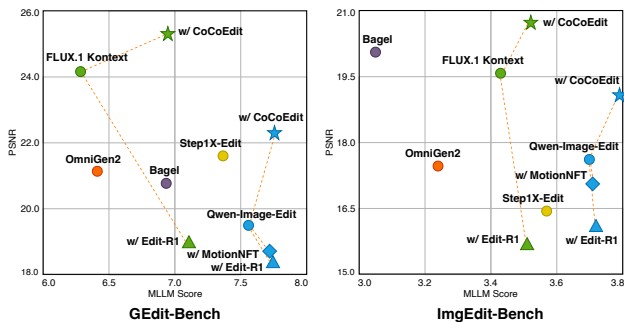

*Figure 1.* Performance comparison on GEdit-Bench and ImgEdit-Bench. The PSNR values are calculated on the non-edit region. The results illustrate a clear conflict between editing quality (MLLM Score) and content consistency (PSNR) of existing editing models. Notably, with our CoCoEdit, both the editing quality and content consistency can be improved.

et al., 2025) and expert editing models (Labs et al., 2025; Wu et al., 2025a), have achieved remarkable success, which can generally understand and follow user instructions faithfully to achieve desired editing goals. However, while these models show strong editing performance on intended objects and regions, they tend to generate unwanted changes in unintended regions, leading to low content consistency. In Figure 1, we illustrate the performance of representative large editing models, including FLUX.1 Kontext (Labs et al., 2025) and Qwen-Image-Edit (Wu et al., 2025a), on GEdit-Bench and ImgEdit-Bench. One can see that there is a clear conflict between editing quality (MLLM Score) and content consistency (PSNR) of existing editing models. Figure 2 shows some visual examples. We see that models such as Qwen-Image-Edit can achieve superior editing effects, but they often change the objects/regions where we want to preserve without modification.

Attempts have been made to apply reinforcement learning (RL) (Liu et al., 2025a) to image editing (Li et al., 2025; Tian et al., 2025), aiming to learn better edits through feedback of the Multimodal Large Language Model (MLLM) (Bai et al., 2025). Nevertheless, neither the original MLLMs nor their fine-tuned variants (Luo et al., 2025) can discriminate fine-grained image consistency in editing. As shown in Figure 1, current RL-based editing methods (Li et al., 2025; Tian et al., 2025) tend to prioritize editing capability but ignore content preservation, resulting in even larger imbalance between

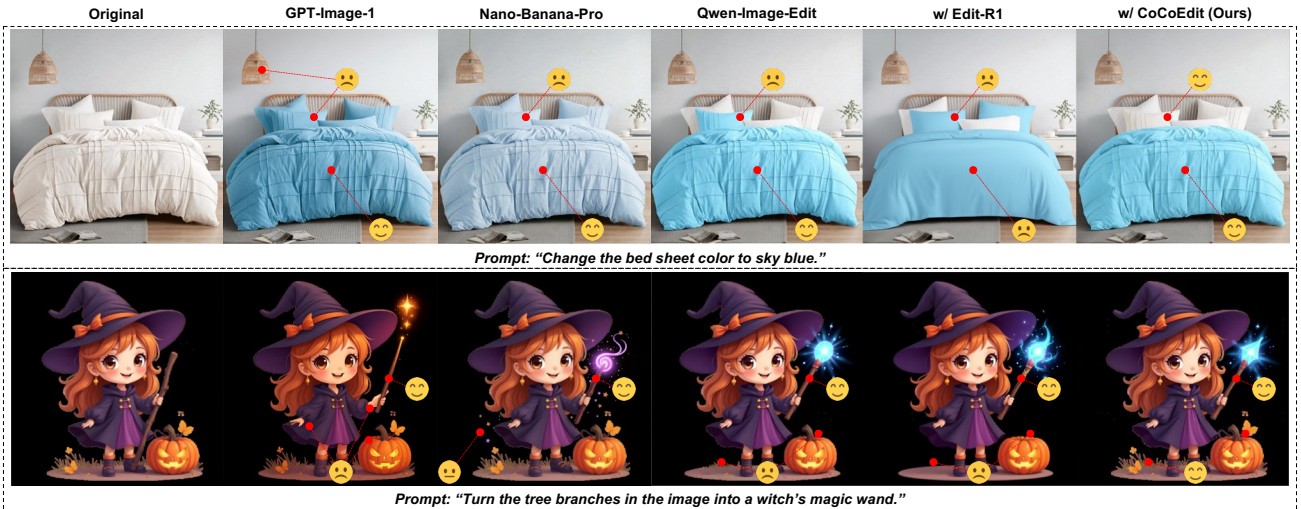

*Figure 2.* Visual examples of state-of-the-art image editing models. We see that models with strong editing capabilities may fail to preserve non-edited objects (*e.g.*, the pillow in the first row), and existing post-training methods can further degrade the consistency of original content. In contrast, our CoCoEdit can improve both editing effects and content consistency.

editing effects and content consistency.

In this work, we propose a post-training framework of **Co**ntent-**Co**nsistent **Edit**ing (**CoCoEdit**), aiming not only to refine the editing effects but also to ensure the content consistency. To this end, we first construct a dedicated training dataset of 40K samples, namely **CoCoEdit-40K**. To enhance the preservation of non-edit regions, we prefer training samples with local editing instructions and the mask of the editing objects. Thus, we collect the data of local editing types from existing datasets (Wei et al., 2024; Ye et al., 2025) with high resolution and diversity, then utilize off-the-shelf large models (Bai et al., 2025; Ravi et al., 2024) to refine the instruction, annotate the masks and filter out high-quality training samples. In addition, we produce segment masks of the editing targets and expand them through dilation to ensure complete coverage of the editing region.

With CoCoEdit-40K, we introduce a pixel-level similarity reward, which consists of normalized PSNR and SSIM scores calculated between the masked regions of sampling output and reference input, to evaluate the changes in the non-edit regions. We utilize this reward to complement MLLM-based scores, enabling the editing model to be aware of editing quality and content consistency simultaneously. However, the pixel-level similarity reward, as a scalar, encourages global consistency but cannot impose precise regional constraints in training. Therefore, we propose region-based regularizers within the objective function to achieve region-aware penalty. Building upon DiffusionNFT (Zheng et al., 2025), which computes diffusion loss directly on image latents, we harness spatial masks to decouple constraints for edit and non-edit regions. Furthermore, we tailor different regularizers for positive and negative samples according to the reward signal. Specifically, for high-reward (positive) samples—where the editing intent is successfully real-

ized—our primary goal is to preserve the non-edit contents. Thus, a positive regularizer is used to enforce consistency within the non-edit regions. However, relying solely on consistency constraints often drives the policy model toward conservative behavior, resulting in insufficient editing and lower rewards. To mitigate this issue, we introduce a negative regularizer for low-reward (negative) samples characterized by under-editing, which is applied to the edit region to explicitly penalize insufficient edits. As illustrated in Figure 2, by using the region regularized RL strategy, our CoCoEdit achieves improved editing effects and content consistency simultaneously.

We evaluate our CoCoEdit method on widely used benchmarks, including GEdit-Bench (Liu et al., 2025b) and ImgEdit-Bench (Ye et al., 2025). However, these benchmarks rely solely on MLLM-based scoring, lacking the capability of pixel-level evaluation. To address this problem, we augment GEdit-Bench and ImgEdit-Bench by annotating masks for the editing regions, introducing pixel-level similarity metrics (PSNR and SSIM), alongside human evaluation, to provide more rigorous assessments on the editing performance. The results on the enhanced benchmarks reveal that while content preservation remains a conspicuous challenge for existing generative editing models, our CoCoEdit achieves substantial improvements.

Our contributions can be summarized as follows. First, we construct CoCoEdit-40K, a high-quality dataset that comprises triplets of inputs, edit masks, and instructions, filtered to ensure mask quality and editing diversity. Second, we present CoCoEdit, a post-training framework equipped with pixel-level similarity rewards and region-based regularizers to achieve superior content-consistent editing. Finally, we augment existing benchmarks with mask annotations to systematically evaluate the performance of both editing

effectiveness and content consistency.

## 2. Related Work

**Instruction-based Image Editing**. By finetuning large diffusion-based image generation models on curated editing datasets (Brooks et al., 2023; Sheynin et al., 2024), instructional image editing has achieved impressive performance (Brooks et al., 2023; Yu et al., 2025; Zhang et al., 2025). Significant progress has also been achieved on data construction pipelines (Zhang et al., 2023; Yu et al., 2025; Ye et al., 2025), including various editing types, higher resolution, and better instructions. To handle complex scenarios, MLLM (Huang et al., 2024; Fu et al., 2023; Fang et al., 2025) have been used to understand user instructions and produce extra information for diffusion models. Recently, unified methods, including BAGEL (Deng et al., 2025), Ovis-U1 (Wang et al., 2025a), and OmniGen2 (Wu et al., 2025b), have also been developed to perform image editing. Specialized editing systems, such as Step1X-Edit (Liu et al., 2025b), FLUX.1 Kontext (Labs et al., 2025), SeedEdit 3.0 (Wang et al., 2025b), Qwen-Image-Edit (Wu et al., 2025a) and LongCat-Image (Team et al., 2025), exhibit strong instruction-based editing capabilities for their large-scale datasets and elaborate training strategies.

**Reinforcement Learning in Image Generation**. Recent research has explored various optimization paradigms to align generative models with human preferences. DPO (Rafailov et al., 2023) eliminates explicit reward models, yet its requirement of offline data is inefficient for large-scale iterative training. PPO (Ren et al., 2024) incorporates online feedback in the adaptation of diffusion models, but it suffers from high computational complexity and training instability. GRPO (Shao et al., 2024) removes the value function to enhance efficiency and has been successfully extended to visual generation. Specifically, Flow-GRPO (Liu et al., 2025a) adapts GRPO to flow matching models by reinforcing optimal trajectories based on reward feedback, while DanceGRPO (Xue et al., 2025) reformulates the sampling of diffusion models and rectified flows for stable training. Unlike these paradigms, DiffusionNFT (Zheng et al., 2025), based on the flow-matching objective, bypasses the policy gradient framework by contrasting positive and negative samples to implicitly integrate reinforcement guidance into the forward diffusion process.

**Reinforcement Learning in Image Editing**. RL techniques have also been used to improve the capability of editing models. Skywork UniPic 2.0 (Wei et al., 2025) presents a unified model capable of understanding, generation, and editing through post-training. UniGen-1.5 (Tian et al., 2025) is also a unified model, which employs shared rewards and light edit instruction alignment for stable RL training. EditScore (Luo et al., 2025) introduces specialized reward mod-

els to evaluate the quality of image editing, and fine-tunes OmniGen2 (Wu et al., 2025b) via RL training. Notably, UniWorld-V2 (Li et al., 2025) exploits DiffusionNFT for image editing and proposes Edit-R1 framework, building a useful code base. MotionEdit (Wan et al., 2025) improves Edit-R1 by using motion alignment rewards, guiding the models towards accurate motion transformations. However, most current RL-based methods focus only on improving editing effects while ignoring the preservation of image contents, including unwanted changes in unintended regions, original layouts, and textures.

## 3. Methods

### 3.1. Dataset Construction

CoCoEdit aims to improve the content consistency of non-edit regions while maintaining the editing performance of edited regions. To achieve this goal, we propose CoCoEdit-40K, an elaborated post-training dataset consisting of triplets of carefully aligned input images, masks, and instructions, to enhance the capability of content-consistent editing. Unlike UniWorld-V2 (Li et al., 2025), which curates text-image pairs from image generation datasets, and MotionEdit (Wan et al., 2025), which collects frame pairs from videos to learn motion editing, we leverage existing image editing datasets whose image pairs and carefully designed instructions are directly accessible. In particular, we collect local editing samples from OmniEdit (Wei et al., 2024) and ImgEdit (Ye et al., 2025) as our original set, since local editing has explicit editing targets, which can provide regional guidance in training process. With these original samples, we develop a data construction pipeline, which consists of three stages (as shown in Figure 3), to refine and filter the training dataset.

**Mask Annotation.** Given the original samples, we first annotate the editing masks using Qwen2.5-VL-72B (Bai et al., 2025) and SAM 2 (Ravi et al., 2024), generating masks of the whole OmniEdit dataset and those samples without masks in the ImgEdit dataset.

**Instruction and Mask Augmentation.** The annotated masks are then used as conditions to refine the original instructions. We use MLLM to supplement simple instructions by injecting spatial positioning and object attributes based on the mask. This refinement explicitly helps the model learn the semantic mapping between instruction, edit region, and non-edit region, which is crucial for content-consistent editing. For the editing type of object addition, we also input the edited images to align with the masks and instructions. The refined instructions are more precisely aligned with the original image and the editing mask, benefiting the region-based rewards and constraints in the CoCoEdit training process. Moreover, some editing types, *e.g.*, replace and motion, often introduce contents with different shapes from the edited targets. Thus, we dilate the

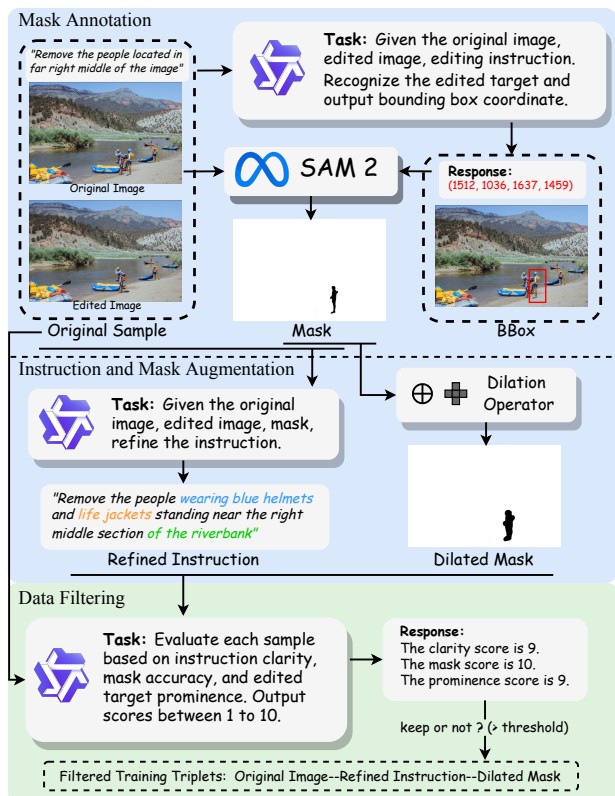

*Figure 3.* Data construction pipeline of our CoCoEdit-40K. In the blue part, we annotate the editing masks, augment them and refine the instruction. In the green part, we filter the augmented samples using Qwen2.5-VL based on the alignment quality between instructions and inputs, and the accuracy of masks.

masks to expand the editing region, ensuring that newly generated contents can be covered as complete as possible.

**Data Filtering.** The refined instructions, the dilated masks, and the original samples are evaluated and filtered by Qwen2.5-VL-72B, as shown in the green part of Figure 3. Different from the filtering protocol used in previous editing datasets (Wei et al., 2024; Ye et al., 2025), which mainly focus on the editing effects, we evaluate the instruction clarity, mask accuracy, and target prominence to ensure the guidance of region consistency. We focus more on the quality of the condition signals (i.e., text and mask) to ensure correct regional guidance during training. Finally, we keep the samples with higher scores, and each sample consists of the original image, the refined instruction, and the dilated mask. Since RL training learns from the generated samples via reward feedback, it eliminates the costly need to annotate high-quality edited images as GT.

Figure 4 summarizes the composition and statistics of our CoCoEdit-40K dataset, which comprises 6 local editing tasks with explicit edit regions: Replace, Remove, Add, Attribute, Background, and Motion. Although we do not include global editing tasks in our dataset, the finetuned

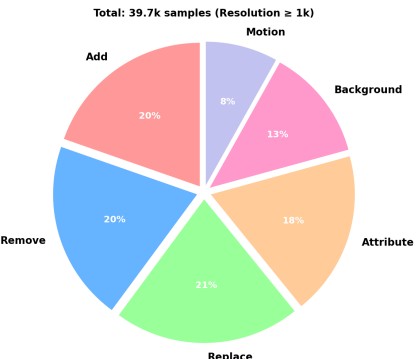

*Figure 4.* Statistics of CoCoEdit-40K. We only consider local editing types for CoCoEdit training, while the finetuned models show generalization capability to global editing types.

models also demonstrate competitive performance on these tasks. Different from existing fine-grained editing methods such as FireEdit (Zhou et al., 2025), which requires explicit spatial grounding at inference time, CoCoEdit **only uses masks as spatial guidance during training**, and **is completely mask-free in inference**, ensuring a fair comparison with baselines. Detailed prompts and dataset statistics are provided in **Appendix A**.

### 3.2. CoCoEdit Training

With the constructed CoCoEdit-40K dataset, we train our CoCoEdit model. As shown in Figure 5, each iteration of CoCoEdit training consists of three stages, including sample collection, reward collection, and policy optimization.

**Sample Collection.** Given the input image $\hat{c}_I$ and the instruction $\hat{c}_T$, a group of edited samples $\hat{x}_0^{1:N}$ is sampled from the old policy $v^{old}$, which is input to the reward function to obtain the overall reward $r$. Samples $\hat{x}_0^{1:N}$ will also be used to generate noisy inputs for policy optimization.

**Reward Functions.** As in existing works (Luo et al., 2025; Li et al., 2025), we employ MLLM as one reward model to compute rewards $r_{mllm}^{1:N}$. Commonly used MLLM-based reward models (Bai et al., 2025; Luo et al., 2025), however, often struggle to distinguish fine-grained changes between images, since they directly evaluate the whole edited image against the instructions to judge the editing effect and do not use any masks. As shown in Figure 5, the different postures in edited samples cannot be identified by MLLM, which yields the same score. Based on our CoCoEdit-40K with aligned editing masks, we propose a pixel-level similarity reward $r_{sim}$ to evaluate the changes in the non-edit region of each sample. Given the input image $\hat{c}_I$, the output samples $\hat{x}_0^{1:N}$ and the full-resolution mask $m \in \{0,1\}^{H \times W}$, we compute the PSNR and SSIM scores between the non-edit regions of $\hat{c}_I$ and $\hat{x}_0^{1:N}$, which are defined as $\text{PSNR}_m(\cdot)$ and $\text{SSIM}_m(\cdot)$. These similarity scores are then normalized into the same scale and averaged as the pixel-level similarity

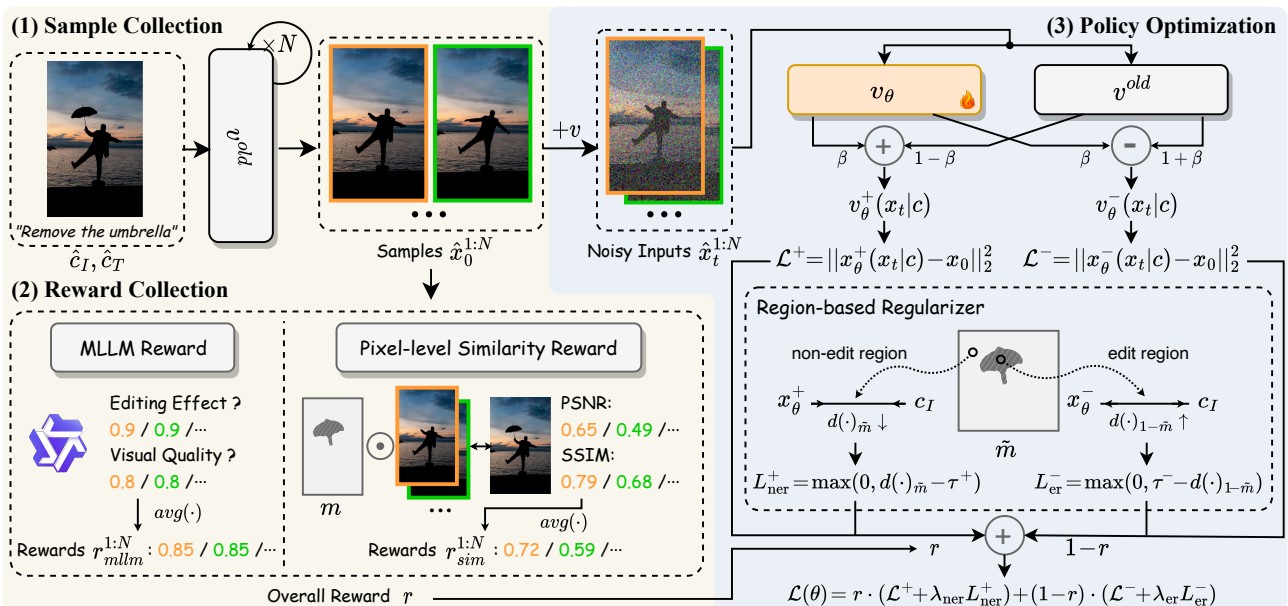

*Figure 5.* Illustration of our **CoCoEdit** framework, which consists of three stages in each iteration. **(1) Sample Collection**. Given the input image and instruction, we collect multiple samples for reward calculation. **(2) Reward Collection.** In addition to the MLLM reward, we propose a pixel-level similarity reward that computes PSNR and SSIM in the non-edit region to identify the inconspicuous differences ignored by MLLM. **(3) Policy Optimization.** The current policy model $v_\theta$ is trained by the negative-aware loss terms $\mathcal{L}^+, \mathcal{L}^-$ and our region-based regularizers $L_{ner}^+, L_{er}^-$, which are weighted by the reward signal $r$, where $L_{ner}^+$ aims to constrain the similarity in non-edit region and $L_{er}^-$ promotes the editing effects in the edit region.

reward $r_{sim}^{1:N}$ as follows:

$$r_{sim}^{1:N} = \{\text{PSNR}_m(\hat{c}_I, \hat{x}_0)), \text{SSIM}_m(\hat{c}_I, \hat{x}_0)\}_{\text{avg}}^{1:N}, \quad (1)$$

where PSNR is normalized into $[0, 1]$ to match the scale of SSIM. For MLLM reward, we use Qwen2.5-VL-32B (Bai et al., 2025) to evaluate the editing effect and visual quality of each sample, yielding the rewards $r_{mllm}^{1:N}$.

Finally, we obtain the overall reward $r$ as:

$$r = \text{op}(\lambda_{mllm} \, r_{mllm}^{1:N} + \lambda_{sim} \, r_{sim}^{1:N}), \quad (2)$$

where $\lambda_{sim}$ and $\lambda_{mllm}$ are weights, $\text{op}(\cdot)$ denotes the transformation of optimality probability (Zheng et al., 2025).

**Policy Optimization.** After collecting the overall reward $r$ and samples $\hat{x}_0^{1:N}$, we update the current diffusion model $v_\theta$ of policy $\pi_\theta$ from the old policy $\pi^{\text{old}}$ in the final stage. We produce noisy inputs $\hat{x}_t^{1:N}$ through forward diffusion, which are input to $v_\theta$ and $v^{\text{old}}$ for velocity prediction. Inspired by NFT (Zheng et al., 2025), the implicit positive policy $v_\theta^+(x_t|c) = (1 - \beta) \, v^{\text{old}}(x_t|c) + \beta \, v_\theta(x_t|c)$ and implicit negative policy $v_\theta^-(x_t|c) = (1 + \beta) \, v^{\text{old}}(x_t|c) - \beta \, v_\theta(x_t|c)$ are derived with a hyperparameter $\beta$. In implementation, the $v$-prediction $v_\theta$ can be transformed into $x$-prediction $x_\theta$ and the loss terms are formulated as $\mathcal{L}^+ = \|x_\theta^+(x_t|c) - x_0\|_2^2$ and $\mathcal{L}^- = \|x_\theta^-(x_t|c) - x_0\|_2^2$. The positive and negative policies are weighted by the reward $r$ to formulate the initial training objective as:

$$\mathcal{L}(\theta) = \mathbb{E}_{c, \pi^{\text{old}}(x_0|c), t} \left[ r \cdot \mathcal{L}^+ + (1 - r) \cdot \mathcal{L}^- \right]. \quad (3)$$

However, the pixel-level similarity reward $r_{sim}^{1:N}$ only provides global consistency feedback for each sample, which limits the learning efficiency for content-consistent editing. The $x$-prediction applies local constraints to different regions of the output latent $x_\theta$. Therefore, we propose region-based regularizers $L_{ner}^+$ and $L_{er}^-$ within the objective function, as shown in the blue part of Figure 5, to achieve region-aware penalty by utilizing the masks to decouple constraints for edit and non-edit regions. $L_{ner}^+$ minimizes the distance of non-edit regions between $x_\theta^+$ and $c_I$, and $L_{er}^-$ increases the distance of edit regions.

Specifically, given the output latent $x_\theta(x_t|c)$, latent of output sample $x_0$, latent of reference input $c_I$, we downsample the spatial dimension of editing mask $m \in \{0, 1\}^{H \times W}$ to obtain $\tilde{m}$, which aligns with the latent shape. Then, we define projection operators $P_{\text{er}}(z) = z \odot (1 - \tilde{m})$ and $P_{\text{ner}}(z) = z \odot \tilde{m}$ to divide the latent features into edit and non-edit regions, respectively. Based on the spatial decoupling of the latents, we formulate distinct regularization terms for positive (high-reward) and negative (low-reward) samples to guide the optimization process precisely. For positive samples $x_\theta^+(x_t|c)$ with which the editing intent is well-aligned, we maintain the content consistency of the non-edit regions. Thus, we define the $L_2$ distance between $P_{\text{ner}}(x_\theta^+(x_t|c))$ and $P_{\text{ner}}(c_I)$ as $d(x_\theta^+(x_t|c), c_I)_{\tilde{m}}$, and introduce the regularizer $L_{\text{ner}}^+$ for positive samples:

$$L_{\text{ner}}^+ = \max(0, d(x_\theta^+(x_t|c), c_I)_{\tilde{m}} - \tau^+), \quad (4)$$

| Model | Background | Color | Material | Motion | Ps_human | Add | Remove | Replace | Text | Overall ↑ | PSNR ↑ | SSIM ↑ | LPIPS ↓ | DINO ↑ | Rank ↓ |
|---|---|---|---|---|---|---|---|---|---|---|---|---|---|---|---|
| BAGEL | 6.843 | 6.629 | 6.900 | 6.505 | 6.345 | 7.515 | 7.386 | 7.238 | 7.099 | 6.940 | 20.783 | 0.745 | 0.203 | 0.819 | - |
| OmniGen2 | 7.330 | 7.133 | 5.798 | 5.875 | 5.237 | 6.813 | 7.113 | 7.033 | 5.361 | 6.410 | 21.116 | 0.736 | 0.197 | 0.824 | - |
| Step1X-Edit | 7.525 | 7.497 | 7.352 | 7.139 | 7.030 | 7.565 | 7.763 | 7.654 | 6.890 | 7.379 | 21.588 | 0.750 | 0.189 | 0.837 | - |
| FLUX.1 Kontext | 7.153 | 6.908 | 5.764 | 5.196 | 5.330 | 7.167 | 6.728 | 6.717 | 5.612 | 6.286 | 24.168 | 0.825 | 0.150 | 0.871 | 2.1 |
| w/ Edit-R1 | 7.335 | 7.499 | 7.093 | 6.664 | 6.790 | 7.497 | 7.269 | 7.110 | 6.758 | 7.113 | 19.013 | 0.716 | 0.214 | 0.804 | 2.6 |
| w/ CoCoEdit (Ours) | 7.144 | 7.228 | 6.647 | 6.615 | 6.010 | 7.784 | 7.252 | 7.001 | 6.772 | 6.939 | **25.331** | **0.874** | **0.139** | **0.882** | 1.6 |
| Qwen-Image-Edit | 7.894 | 7.588 | 6.990 | 7.359 | 6.776 | 7.836 | 7.783 | 7.865 | 7.945 | 7.560 | 19.488 | 0.662 | 0.185 | 0.831 | 2.7 |
| w/ Edit-R1 | 7.877 | **7.733** | **7.459** | 7.609 | **7.251** | 7.832 | 7.906 | 7.876 | 8.175 | 7.746 | 18.441 | 0.639 | 0.214 | 0.804 | 3.3 |
| w/ MotionNFT | 7.880 | 7.705 | 7.267 | **7.760** | 7.172 | 7.828 | 7.747 | 7.864 | 8.179 | 7.711 | 18.709 | 0.642 | 0.201 | 0.813 | 2.9 |
| w/ CoCoEdit (Ours) | **7.898** | 7.675 | 7.329 | 7.758 | 7.001 | **7.931** | **7.975** | 7.876 | **8.331** | **7.754** | 22.283 | 0.774 | 0.162 | 0.852 | **1.4** |

Table 1. Quantitative comparison with existing methods on GEdit-Bench-EN. The best results are highlighted in **bold**.

| Model | Add | Adjust | Replace | Remove | Background | Compose | Action | Overall ↑ | PSNR ↑ | SSIM ↑ | LPIPS ↓ | DINO ↑ | Rank ↓ |
|---|---|---|---|---|---|---|---|---|---|---|---|---|---|
| BAGEL | 3.27 | 3.10 | 3.21 | 3.10 | 2.86 | 2.62 | 3.67 | 3.12 | 20.062 | 0.611 | 0.318 | 0.729 | - |
| OmniGen2 | 3.22 | 3.37 | 3.24 | 3.18 | 3.34 | 2.59 | 3.74 | 3.24 | 17.482 | 0.578 | 0.364 | 0.668 | - |
| Step1X-Edit | 3.48 | 3.75 | 3.55 | 4.15 | 3.40 | 2.87 | 3.79 | 3.57 | 16.471 | 0.468 | 0.369 | 0.665 | - |
| FLUX.1 Kontext | 3.53 | 3.60 | 3.60 | 3.31 | 3.36 | 2.80 | 3.78 | 3.43 | 19.591 | 0.592 | 0.303 | 0.735 | 2.2 |
| w/ Edit-R1 | 3.58 | 3.55 | 3.63 | 3.63 | 3.52 | 2.94 | 3.74 | 3.51 | 15.722 | 0.506 | 0.356 | 0.670 | 2.5 |
| w/ CoCoEdit (Ours) | 3.64 | 3.53 | 3.57 | 3.85 | 3.33 | 2.82 | **3.87** | 3.52 | **20.747** | **0.635** | **0.297** | **0.743** | **1.5** |
| Qwen-Image-Edit | 3.80 | 3.77 | 3.66 | 4.33 | 3.63 | 2.92 | 3.78 | 3.70 | 17.635 | 0.526 | 0.359 | 0.672 | 2.7 |
| w/ Edit-R1 | 3.74 | 3.70 | 3.73 | 4.39 | 3.71 | **3.01** | 3.79 | 3.72 | 16.130 | 0.482 | 0.394 | 0.634 | 3.1 |
| w/ MotionNFT | 3.83 | 3.75 | 3.67 | 4.29 | 3.70 | 2.95 | 3.81 | 3.71 | 17.072 | 0.514 | 0.369 | 0.660 | 2.8 |
| w/ CoCoEdit (Ours) | **3.85** | **3.96** | **3.76** | **4.47** | **3.74** | 2.87 | 3.85 | **3.79** | 19.125 | 0.555 | 0.331 | 0.694 | 1.7 |

Table 2. Quantitative comparison with existing methods on ImgEdit-Bench. The best results are highlighted in **bold**.

where we set $\tau^+$ to realize the regularization hinge. $L_{\text{ner}}^+(\theta)$ penalizes deviations in the non-edit regions that exceed the tolerance threshold $\tau^+$, resulting in more consistent outputs. However, we find that rely solely on $L_{\text{ner}}^+(\theta)$ often incorrectly push the policy model toward insufficient editing effects. To mitigate this issue, we introduce a negative regularizer $L_{\text{er}}^-(\theta)$ for low-reward samples that often suffer from under-editing. We define the $L_2$ distance between $P_{\text{er}}(x_\theta^-(x_t|c))$ and $P_{\text{er}}(c_I)$ as $d(x_\theta^-(x_t|c), c_I)_{1-\tilde{m}}$. $L_{\text{er}}^-$ aims to penalize the insufficient changes in the edit regions as follows:

$$L_{\text{er}}^- = \max(0, \tau^- - d(x_\theta^-(x_t|c), c_I)_{1-\tilde{m}}), \quad (5)$$

where $\tau^-$ is an adaptive threshold to fit various editing types. This term explicitly encourages the model to generate structural modifications if the difference is smaller than a margin $\tau^-$, thereby promoting more edit effects. The detailed analysis of $\tau$ is shown in appendix B.3. Combining $L_{\text{ner}}^+$, $L_{\text{er}}^-$ with the standard diffusion losses $\mathcal{L}^+, \mathcal{L}^-$ in Equation (3), we have the overall objective $\mathcal{L}(\theta)$:

$$\mathcal{L}(\theta) = \mathbb{E}_{c,\pi^{\text{old}}(x_0|c),t}\Big[r \cdot (\mathcal{L}^+ + \lambda_{\text{ner}}L_{\text{ner}}^+)$$
$$+ (1-r) \cdot (\mathcal{L}^- + \lambda_{\text{er}}L_{\text{er}}^-)\Big], \quad (6)$$

where $\lambda_{\text{ner}}$ and $\lambda_{\text{er}}$ are hyperparameters balancing the regularization strength. This formulation ensures that the policy model can obtain precise regional feedback, learn to preserve the content from successfully edited samples, and encourage editing effects for under-edit samples. The pseudo code and analysis of CoCoEdit are shown in **Appendix B**.

### 3.3. Content Consistency Assessment

Recent popular benchmarks, *e.g.*, GEdit-Bench (Liu et al., 2025b) and ImgEdit-Bench (Ye et al., 2025), evaluate the editing performance via MLLMs, which are not able to assess content consistency. Instead of constructing a new benchmark, we extend GEdit-Bench and ImgEdit-Bench by annotating editing masks and incorporating consistency-aware metrics, *i.e.*, PSNR, SSIM, to address this limitation. This extension maintains compatibility with existing MLLM scores, facilitating a direct and fair comparison with state-of-the-art methods. As shown in the blue part of Figure 3, we adopt Qwen2.5-VL-72B (Bai et al., 2025) to output the bounding box of the editing target and obtain the mask through SAM 2 (Ravi et al., 2024), which is then dilated. Then, we check the samples and refine the wrong masks manually to ensure the quality of masks. Finally, the benchmarks are capable of calculating PSNR and SSIM in non-edit regions with the masks. To ensure the rationality of mask generation, we exclude unsuitable editing types such as global editing (style, tone), location-agnostic editing (addition) and layout change (extract).

## 4. Experiments

### 4.1. Experimental Settings

**Implementation Details.** We train our CoCoEdit models by finetuning FLUX.1 Kontext [Dev] (Labs et al., 2025) and Qwen-Image-Edit [2509] (Wu et al., 2025a) using LoRA (Hu et al., 2022) with rank of 32. We perform training with a node of 8 NVIDIA A800 GPUs for 1K iterations with

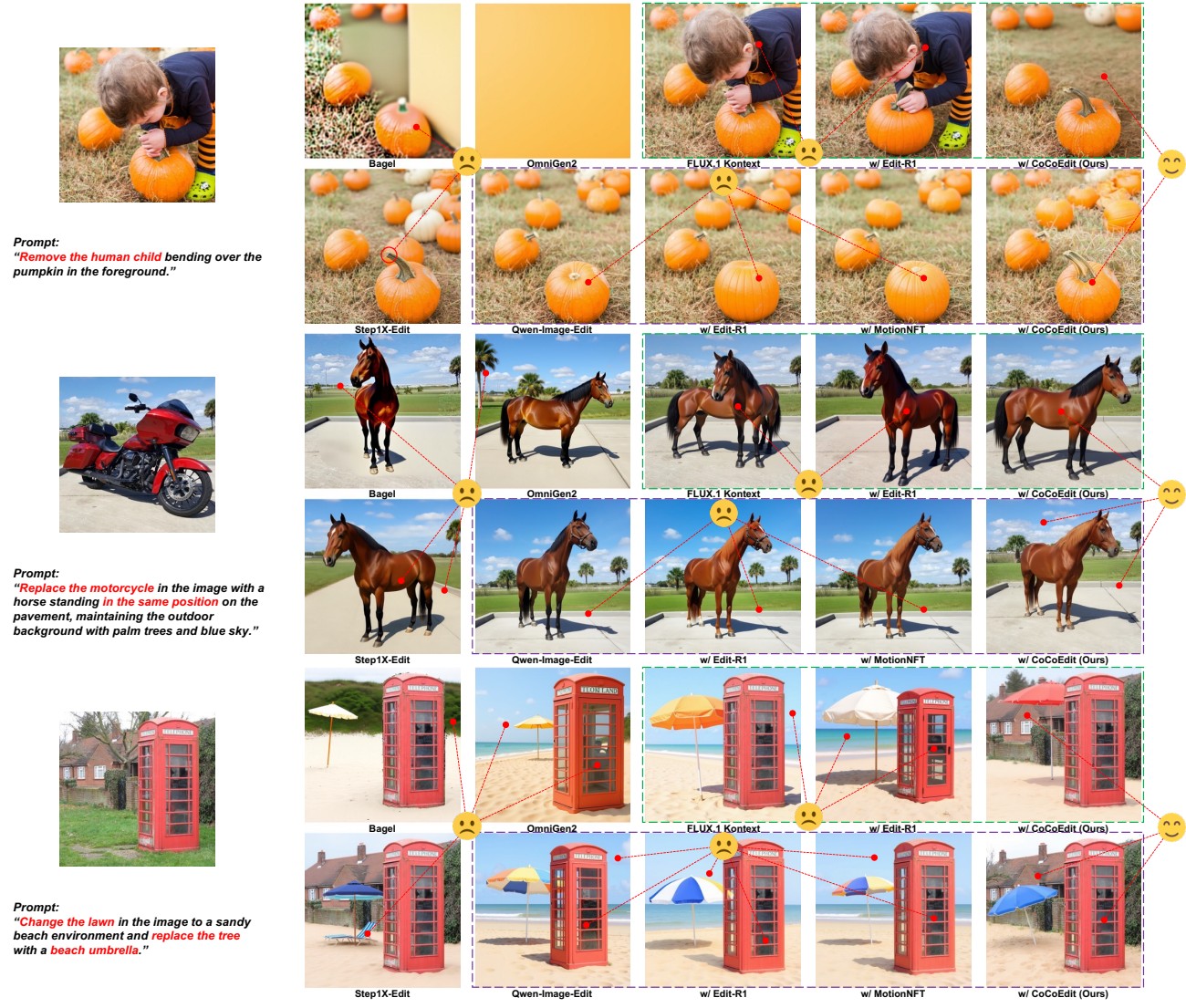

*Figure 6.* Visual comparison between our CoCoEdit and state-of-the-art methods. The baselines fail to produce semantically aligned editing effects and preserve non-edit contents simultaneously. Edit-R1 and MotionNFT still remain the shortage of consistency, while our CoCoEdit successfully promotes the content-consistent editing capability of the base models.

batch size of 3 and group size of 12. For reward functions, the MLLM reward is obtained by deploying Qwen2.5-VL-32B (Bai et al., 2025) on another node as a remote server, and the pixel-level similarity reward is computed on the training node. More details are shown in **Appendix C**.

**Compared Methods.** Besides the baseline models FLUX.1 Kontext [Dev] and Qwen-Image-Edit [2509], we compare with several representative works, including unified multimodal models BAGEL (Deng et al., 2025) and Omni-Gen2 (Wu et al., 2025b), specialized editing model Step1X-Edit (Liu et al., 2025b), and post-training frameworks Edit-R1 (Li et al., 2025) and MotionNFT (Wan et al., 2025).

**Evaluation.** We evaluate the compared methods on ImgEdit (Ye et al., 2025) and GEdit-Bench (Liu et al.,

2025b) with mask annotations. The MLLM evaluation protocols are conducted using Qwen2.5-VL-72B (Bai et al., 2025). For the set of local editing types, we evaluate the content consistency with annotated masks. Specifically, we compute PSNR and SSIM in non-edit regions to obtain the pixel-level similarity between input and edited images. We also conduct a user study on 100 randomly selected samples for a more comprehensive evaluation. The details of user study are presented in **Appendix C.2**.

### 4.2. Quantitative Results

As shown in Table 1 and Table 2, we first make quantitative comparisons between CoCoEdit and state-of-the-art methods on local editing tasks, for which content con-

sistency is crucial. It can be clearly seen that CoCoEdit achieves a significant improvement over its baseline models (FLUX.1 Kontext or Qwen-Image-Edit) in consistency metrics (PSNR and SSIM) while improving their editing scores. For example, on GEdit-Bench-EN, CoCoEdit improves the PSNR of Qwen-Image-Edit and FLUX.1 Kontext by 2.8dB and 1.16dB, respectively, and improves their overall editing scores by 0.19 and 0.65, respectively. In comparison, existing RL-based approaches such as Edit-R1 and MotionNFT can only improve the editing scores but result in severe degradation in content preservation. For example, on GEdit-Bench-EN, Edit-R1 reduces the PSNR of the baseline models FLUX.1 Kontext and Qwen-Image-Edit by 5.15dB and 1.04dB, respectively. The significant improvements in PSNR and SSIM over the baselines are fundamentally the result of the enhanced consistency achieved by our CoCoEdit training framework. The high PSNR and SSIM scores reflect actual background preservation, rather than merely overfitting to the reward metrics. These results demonstrate that CoCoEdit addresses well the conflict between editing effects and content consistency. Furthermore, we evaluate our method and baselines using LPIPS and DINO structural distance. **Note that neither of them is used as the training reward**. CoCoEdit still demonstrates a significant advantage on these metrics, confirming that our method genuinely improves the content consistency of the edited results. Overall, with FLUX.1 Kontext as the baseline model, CoCoEdit achieves the highest PSNR/SSIM/LPIPS/DINO scores with competitive overall editing scores on both GEdit-Bench-EN and ImgEdit-Bench; with Qwen-Image-Edit as the baseline model, CoCoEdit achieves the highest overall editing scores with highly competitive PSNR/SSIM/LPIPS/DINO scores.

In the right column of Table 1 and Table 2, we show the human ranking of FLUX.1 Kontext and Qwen-Image-Edit and its RL-based variants (Edit-R1, MotionNFT and CoCoEdit), respectively. (Please refer to **Appendix C.2** for the details of human subjective evaluation.) We see that CoCoEdit significantly outperforms its competitors in human preference, supporting the superiority of its objective results.

Although CoCoEdit is trained on local editing samples, it does not compromise global editing capabilities. Table 3 compares the global editing performance of the baseline models and its RL-based methods. We see that CoCoEdit improves the base models across most metrics except minor drops on ImgEdit-Bench. (We include the 'Extract' edit type due to the need of global layout change in most cases.) For FLUX.1 Kontext, CoCoEdit provides comparable performance to the baseline model across benchmarks. For Qwen-Image-Edit, CoCoEdit achieves the best scores of Style. The results demonstrate that global editing, especially Style and Tone that require structural preservation, can benefit from the pixel-level consistency training process.

| Model | GEdit-Bench | | ImgEdit-Bench | |
|---|---|---|---|---|
| | Style | Tone | Style | Extract |
| FLUX.1 Kontext | 5.690 | 6.477 | 3.31 | 2.75 |
| w/ Edit-R1 | 6.553 | 6.814 | 3.35 | 2.84 |
| w/ CoCoEdit (Ours) | 5.761 | 6.706 | 3.29 | 2.71 |
| Qwen-Image-Edit | 6.666 | 7.156 | 3.39 | 3.61 |
| w/ Edit-R1 | 6.935 | 7.150 | 3.43 | **3.88** |
| w/ MotionNFT | 6.962 | **7.541** | 3.37 | 3.68 |
| w/ CoCoEdit (Ours) | **6.992** | 7.207 | **3.53** | 3.65 |

*Table 3.* Quantitative comparison of global editing types.

### 4.3. Visual Comparisons

Figure 6 provides visual comparisons between CoCoEdit and its competitors. We see that CoCoEdit improves both FLUX.1 Kontext and Qwen-Image-edit, producing better content consistency, as well as editing effects. To be specific, in the removal task, FLUX.1 Kontext fails to edit, and Edit-R1 produces incorrect details of the child and pumpkin. Qwen-Image-Edit and its variants with Edit-R1 and Motion-NFT cause unexpected changes in the pumpkin area as well. In contrast, CoCoEdit completely deletes the child and preserves the original details. The middle sample with longer instruction brings difficulties to both FLUX.1 Kontext and Qwen-Image-Edit, which synthesize horse with wrong structure or change the background severely. FLUX.1 Kontext with Edit-R1 refines the structure but fails to precisely preserve the background. CoCoEdit follows the editing instruction accurately while keeping the background consistent with the input. For Qwen-Image-Edit, CoCoEdit preserves the non-edit region and maintains the replacement capability, whereas Edit-R1 and MotionNFT suffer from inconsistent background. For the bottom sample, the base models struggle to control the editing effects within the target. They replace the lawn with a beach and extend the beach to the non-edit house. The results of Qwen-Image-Edit's variants also lose details of the telephone booth. Step1X-Edit remains the house and the telephone booth, but adds an unexpected lounge chair. CoCoEdit achieves satisfying editing effects and content preservation for both base models.

### 4.4. Ablation Study

**Reward Ratio.** We ablate the ratio of rewards and the contribution of region-based regularizer on GEdit-Bench in Figure 7. The blue curve, where $\lambda_{mllm}, \lambda_{sim} = 0.5, 0.5$, shows a severe reduction in editing score (left) and a rapid increase in PSNR (right), which finally produces outputs without any editing effect. This is because, in this setting, pixel-level similarity metrics provide more sensitive rewards than the MLLM feedback, and they force the model to get higher reward through more conservative editing. The setting of $\lambda_{mllm}, \lambda_{sim} = 0.8, 0.2$ (orange curve) provides a more stable optimization of the editing score (left) by compromising the upper bound of consistency (right). With region-based regularizer (the green curve), we further achieve an im-

| Setting | GEdit-Bench | | ImgEdit-Bench | |
|---|---|---|---|---|
| | Overall | PSNR | Overall | PSNR |
| Qwen-Image-Edit | 7.560 | 19.488 | 3.70 | 17.635 |
| w/ SFT on 40K | 7.219 | 20.293 | 3.61 | 18.048 |
| w/ RL on 120K | 7.723 | 22.204 | **3.79** | **19.201** |
| w/ RL on 40K (Ours) | **7.754** | **22.283** | **3.79** | 19.125 |

*Table 4.* Effects of dataset size and SFT on GEdit-Bench and ImgEdit-Bench.

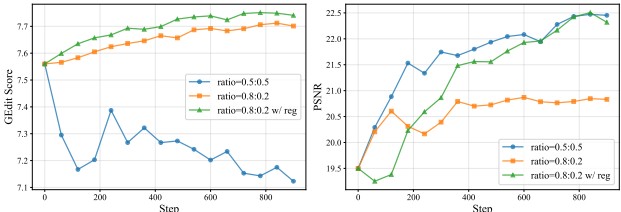

*Figure 7.* Ablated comparison of different ratios between $\lambda_{mllm}$ and $\lambda_{sim}$ and the region-based regularizer on Qwen-Image-Edit.

provement in editing effects and convergence speed. These ablation studies illustrate the effectiveness of our CoCoEdit.

**Impact of Dataset Size.** Unlike SFT, RL training does not require massive datasets to converge, as the model learns from reward-driven exploration rather than fitting ground-truth patterns. Data quality is more important than scale. Prior works like Edit-R1 and MotionEdit also use relatively small datasets (e.g., 27K and 11K). To empirically validate this, we conducted an additional ablation using a larger dataset. As shown in Table 4, scaling up the data size yields little improvement in performance, confirming that our 40K high-quality dataset is sufficient for the RL optimization.

**SFT Training.** To differentiate the contributions of the dataset and the RL algorithm, we extract the GT images of samples in CoCoEdit-40K from source datasets and perform SFT. To make it fair, we add a consistency loss (calculating PSNR/SSIM on the non-edited regions between the output and GT) during training. As shown in Table 4, while the SFT baseline achieves a slight improvement in consistency metrics over the base model due to the explicitly added consistency loss, its Overall editing quality actually drops. This can be expected since our data curation pipeline is specifically tailored for RL and does not filter for the visual quality or instruction following of GT images, which are critical for effective SFT. In contrast, our CoCoEdit significantly outperforms both the base model and the SFT baseline across all metrics. This clearly demonstrates that the substantial gains in both editing quality and consistency stem primarily from our proposed RL training algorithm and reward formulation, rather than merely from the curated data itself.

We provide more discussions in **Appendix C**.

### 4.5. Training time and VRAM usage

We investigate the training time and VRAM usage between CoCoEdit and baseline models for efficiency evaluation. Specifically, since CoCoEdit does not introduce any additional models into the RL training (masks are pre-computed and regularizers are handled via lightweight tensor operations in latent space), the VRAM usage is identical to standard Edit-R1 baseline ($\sim$70GB). Regarding training time, region-based masking and reward calculation add only a marginal latency (Edit-R1 costs $\sim$10 min for one step and CoCoEdit costs $\sim$12 min).

### 5. Conclusion

We proposed a post-training framework of content-consistent editing, namely CoCoEdit, to improve not only the editing effect but also the content consistency. We developed a pixel-level similarity reward to complement the MLLM reward for accurate consistency feedback, and a region-based regularizer to differentiate the edit and non-edit regions for precise regional constraints in training. To facilitate CoCoEdit training, we constructed CoCoEdit-40K, a dedicated dataset of 40K samples with triplets of inputs, edit masks, and instructions. To evaluate CoCoEdit, we annotated GEdit-Bench and ImgEdit-Bench with masks for editing regions and introduced similarity metrics and user studies for a comprehensive evaluation. Both quantitative and qualitative experiments demonstrated that CoCoEdit improved not only significantly the content consistency of state-of-the-art baseline models and their RL-based variants, but also achieved competitive editing effects.

### Impact Statement

This paper introduces a method to improve image editing capabilities. While our goal is to assist creative tasks, we acknowledge that generative models can be misused to create misleading content. We believe our work does not introduce new ethical risks beyond those already known in the field of text-to-image generation, and we support the continued development of safety measures for AI tools.

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

# Appendix

In this appendix, we provide additional materials to supplement the main paper:

- **Section A** provides more details of CoCoEdit-40K, including prompts used to generate coordinates, instruction refinement and filtering, and the score distribution of the original dataset.

- **Section B** provides more details of our CoCoEdit training framework, including a brief preliminary, the calculation of pixel-level reward, analysis of region-based regularizer, and the pseudo code of CoCoEdit.

- **Section C** provides detailed experimental settings and results, including implementation details of CoCoEdit model training and inference, user study, comparison with commercial models, ablation studies, and more visual comparisons.

- **Section D** discusses the failure cases and limitations of CoCoEdit.

## A. Details of CoCoEdit-40K

### A.1. Score Distribution

To quantitatively assess the quality of our constructed dataset, we employ Qwen2.5-VL as an automated evaluator to score the approximately 500K raw samples. As detailed in the filtering pipeline, the scoring mechanism evaluates three pivotal dimensions: instruction clarity, mask accuracy, and target prominence. The final score is the average of the three scores.

Figure 8 illustrates the distribution of these scores on a logarithmic scale. The overall distribution exhibits a significant concentration on the high-score regions, and the majority samples fall into the $[7, 9)$ interval. Specifically, the $[8, 9)$ bin contains the largest volume of data ($\approx$ 199k), indicating that the base generation pipeline produces generally capable samples. However, to construct a high-quality training dataset, we aim to filter out passable samples.

We perform a fine-grained analysis in the high-confidence interval $[9.0, 10.0]$. As observed in the histogram, the sample count decreases gradually as the score approaches to the perfect score of 10. Balancing data scale with alignment quality, we select a strict filtering threshold of **9.4**. We retain only samples with scores exceeding this threshold, effectively filtering out instances with minor imperfections in region consistency or instruction alignment. Consequently, this rigorous selection process yields a final set of **39.7K** high-quality triplets (comprising the original image, refined instruction, and dilated mask) for training, representing about the top $8\%$ of the original dataset.

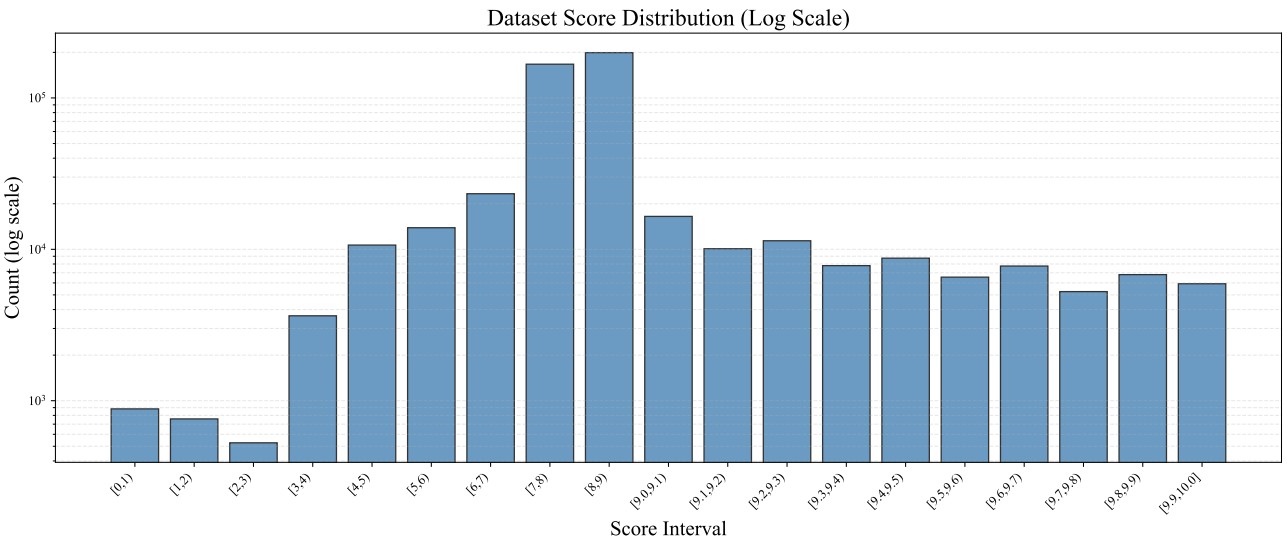

*Figure 8.* Distribution of quality scores for the raw dataset on a logarithmic scale. The intervals are refined for scores above 9.0 to better visualize the high-quality tail.

## A.2. Prompts for Coordinate Generation

In the first stage of data processing, we instruct Qwen2.5-VL to identify the editing type and locate the target object.

---

**System Prompt for Coordinate Generation**

**Role:** You are a multi-modal expert assistant skilled in visual reasoning.
**Task:** You will be given three types of inputs: the original image, the edited image, and an editing instruction. Please reason and output your answers in the following two steps, and finally output all results in strict JSON format.
**Step 1:** According to the editing instruction and the difference between the two images, using verb phrase to describe the editing type based on the given simple editing type '{edit_type}'. There are some examples: add, remove, replace, change color, change appearance/texture/material, change action/motion, change background, change location, etc. And extract the edited object from the editing instruction (can be a noun or noun phrase).
**Step 2:** Based on the identified object, infer its bounding box in either the original or edited image (format: $[x1, y1, x2, y2]$, where $(x1, y1)$ is the top-left corner and $(x2, y2)$ is the bottom-right corner in pixel coordinates. For "add" edits, locate the object in the edited image; for other types, locate it in the original image).
**Output Format:** Finally, output all results in the following strict JSON format (use double quotes for all keys and values):

```
{"caption": "{caption}",
  "edit_type": <verb phrase that describes editing type>,
  "edited_object": "<object or phrase>",
  "bounding_box": [x1, y1, x2, y2]}
```

Make sure the output is strictly valid JSON, with no explanatory text.
**Inputs:** Original Image: <IMAGE_0>, Edited Image: <IMAGE_1>, Editing Instruction: "{caption}"

---

## A.3. Prompt for Instruction Refinement

In the second stage, we utilize the bounding box obtained in the previous step to generate a segmentation mask via SAM2. This mask is then fed into the Qwen2.5-VL as an auxiliary visual cue to generate a refined and detailed editing instruction.

---

**System Prompt for Instruction Refinement**

**Role:** You are a multi-modal expert assistant skilled in editing instruction enhancement.
**Task:** You will be given four inputs: the original image, the edited image, a binary mask highlighting the edited object, and a simple editing instruction. Your goal is to refine the instruction based on the visual details within the masked region.
**Step 1:** Observe the region highlighted by the mask in the images. Analyze the specific changes applied to this object (e.g., texture details, exact color shifts, positional changes, or specific interactions).
**Step 2:** Enhance the original editing instruction by adding more details that describe the object within the mask, such as its spatial position, relative relationships, object attributes, state, color, size, pose, or action.
**Requirements:**
(1) The new instruction should be more detailed and complex. (2) Preserve the original textual structure and key words of the original editing instruction. (3) The new instruction should be concise and clarified.
**Output Format:** Output the result in the following strict JSON format:

```
{"original_caption": "{caption}",
  "complex_editing_instruction": "<the new, more detailed instruction>"}
```

Make sure the output is strictly valid JSON, with no explanatory text.
**Inputs:** Original Image: <IMAGE_0>, Edited Image: <IMAGE_1>, Object Mask: <IMAGE_2>, Editing Instruction: "{caption}"

---

## A.4. Prompts for Filtering

To ensure the high quality of the constructed dataset, we employ a filtering mechanism to evaluate the data samples. We instruct Qwen2.5-VL to assess the samples based on Instruction Clarity, Mask Accuracy, and Target Prominence.

**System Prompt for Data Filtering**

**Role:** You are a multi-modal expert assistant skilled in data quality evaluation.
**Task:** You will be provided with an original image, an edited image, an editing mask, and an editing instruction. Your task is to evaluate the quality of this data sample based on three specific metrics. Provide a score from 1 to 10 for each metric.
**Evaluation Metrics:**
**1. Instruction Clarity (1-10):** Evaluate how clear, specific, and unambiguous the editing instruction is.

- **10:** The instruction is grammatically correct, precise, and leaves no room for ambiguity regarding what needs to be changed.
- **1:** The instruction is nonsense, empty, extremely vague, or grammatically broken to the point of being unintelligible.

**2. Mask Accuracy (1-10):** Evaluate how accurately the binary mask covers the object mentioned in the instruction within the original image.

- **10:** The mask perfectly covers the intended object (and only that object) with precise boundaries.
- **1:** The mask covers the wrong object, the background, or is completely misaligned with the object described in the text.

**3. Target Prominence (1-10):** Evaluate the visual prominence of the target object (the object to be edited) in the original image.

- **10:** The object is the main subject of the image, large, centrally located, and not obscured.
- **1:** The object is tiny, heavily occluded, extremely blurry, or barely visible in the background.

**Output Format:** Output the result in the following strict JSON format (do not include any reasoning text):

```
{"instruction_clarity": <int>,
  "mask_accuracy": <int>,
  "target_prominence": <int>}
```

**Inputs:** Original Image: <IMAGE_0>, Edited Image: <IMAGE_1>, Editing Mask: <IMAGE_2>, Editing Instruction: "{caption}"

## B. More Details of the Method

### B.1. Preliminary

**Flow Matching.** Given a data sample $x_0 \sim X_0$ with a corresponding condition $c$ (e.g., a class label or text embedding), and a Gaussian noise sample $x_1 \sim X_1$, Rectified Flow (Liu et al., 2022; Lipman et al., 2022) defines an interpolated sample $x_t$ as:

$$x_t = (1 - t)x_0 + tx_1, \tag{7}$$

where $t \in [0, 1]$. A neural network $v_\theta(x_t, t, c)$ is trained to approximate the target velocity field $v = x_1 - x_0$ by minimizing the flow matching objective:

$$\mathcal{L}_{\text{FM}}(\theta) = \mathbb{E}_{t, x_0 \sim X_0, x_1 \sim X_1} \left[ \|v - v_\theta(x_t, t, c)\|_2^2 \right]. \tag{8}$$

Inference is performed by solving a deterministic Ordinary Differential Equation (ODE) for the process:

$$dx_t = v_\theta(x_t, t, c)\, dt. \tag{9}$$

**DiffusionNFT.** Based on the flow matching objective described above (often referred to as $v$-prediction), Diffusion-NFT (Zheng et al., 2025) introduces positive velocity $v^+(x_t, t, c)$ and negative velocity $v^-(x_t, t, c)$ to form a bilateral reward signal in post-training:

$$\mathcal{L}_v(\theta) = \mathbb{E}\left[ r \|v_\theta^+ - v\|_2^2 + (1 - r) \|v_\theta^- - v\|_2^2 \right], \tag{10}$$

where $r$ is the normalized reward, and $v_\theta^+, v_\theta^-$ are combinations of the old policy $v^{\text{old}}$ and the current policy $v_\theta$:

$$\begin{aligned}
v_\theta^+(x_t, t, c) &= (1 - \beta)\, v^{\text{old}}(x_t, t, c) + \beta\, v_\theta(x_t, t, c), \\
v_\theta^-(x_t, t, c) &= (1 + \beta)\, v^{\text{old}}(x_t, t, c) - \beta\, v_\theta(x_t, t, c).
\end{aligned} \tag{11}$$

In the implementation, DiffusionNFT transforms the $v$-prediction into $x$-prediction $x_\theta(x_t, t, c)$:

$$\mathcal{L}_x(\theta) = \mathbb{E}\left[ w(t) \|x_\theta(x_t, t, c) - x_0\|_2^2 \right]. \tag{12}$$

This formulation allows us to apply regional constraints to the image latent during training. Theoretically, given unlimited data and model capacity, the optimal solution $v_{\theta*}$ for Eq. (10) satisfies:

$$v_{\theta*}(\boldsymbol{x}_t, \boldsymbol{c}, t) = v^{\text{old}}(\boldsymbol{x}_t, \boldsymbol{c}, t) + \frac{2}{\beta}\Delta(\boldsymbol{x}_t, \boldsymbol{c}, t), \tag{13}$$

which indicates that the optimal update direction is a superposition of the original policy and a scaled reward-driven shift $\Delta$.

### B.2. Detailed Calculation of Pixel-level Similarity Reward

As in prior works (Luo et al., 2025; Li et al., 2025), we employ an MLLM as a primary reward model to compute semantic alignment rewards $r_{\text{mllm}}^{1:N}$. However, commonly used MLLM-based reward models (Bai et al., 2025; Luo et al., 2025) often struggle to distinguish fine-grained pixel-level changes. As illustrated in Figure 5, subtle variations in posture or background preservation in edited samples may not be captured by the MLLM, yielding identical scores for visually distinct outputs.

To address this, we propose a pixel-level similarity reward $r_{\text{sim}}$ to explicitly evaluate the consistency of the non-edit regions. Given the input image $\hat{c}_I$, the generated sample $\hat{x}_0$, and a binary mask $m$ (where $m_{i,j} = 1$ indicates the region to evaluate, typically the background), we calculate the similarity using PSNR and SSIM.

**Masked PSNR.** First, the Masked Mean Squared Error (MSE) is computed as:

$$\text{MSE}_m(\hat{c}_I, \hat{x}_0) = \frac{1}{\sum_{i,j} m_{i,j}} \sum_{i,j} m_{i,j} \left\| \hat{c}_I^{(i,j)} - \hat{x}_0^{(i,j)} \right\|^2. \tag{14}$$

The Masked PSNR is then derived and normalized to the range $[0, 1]$ using a factor $\tau_{\text{db}}$ (set to $40.0$ dB in our implementation, which is high enough in practice):

$$\overline{\text{PSNR}}_m = \text{clip}\left( \frac{10 \cdot \log_{10}(\text{MAX}_I^2/\text{MSE}_m)}{\tau_{\text{db}}}, 0, 1 \right), \tag{15}$$

where $\text{MAX}_I$ is the maximum pixel value.

**Masked SSIM.** To compute the SSIM index strictly within the masked region, we employ a normalized convolution approach. Let $G$ be a Gaussian window with size $11 \times 11$ and $\sigma = 1.5$. We first compute the effective mask area map $M_{\text{area}} = G * m$, where $*$ denotes convolution. The local mean intensities $\mu_I, \mu_x$ and variances $\sigma_I^2, \sigma_x^2, \sigma_{Ix}$ for the reference $\hat{c}_I$ and candidate $\hat{x}_0$ are computed as:

$$\begin{aligned}
\mu_I &= (G * (\hat{c}_I \odot m)) \oslash M_{\text{area}}, \\
\mu_x &= (G * (\hat{x}_0 \odot m)) \oslash M_{\text{area}}, \\
\sigma_I^2 &= (G * (\hat{c}_I^2 \odot m)) \oslash M_{\text{area}} - \mu_I^2, \\
\sigma_{Ix} &= (G * (\hat{c}_I \cdot \hat{x}_0 \odot m)) \oslash M_{\text{area}} - \mu_I \mu_x,
\end{aligned} \tag{16}$$

where $\odot$ and $\oslash$ represent element-wise multiplication and division, respectively. The local SSIM map $S$ is then calculated using the standard formulation:

$$S(p) = \frac{(2\mu_I \mu_x + C_1)(2\sigma_{Ix} + C_2)}{(\mu_I^2 + \mu_x^2 + C_1)(\sigma_I^2 + \sigma_x^2 + C_2)}, \tag{17}$$

where $C_1$ and $C_2$ are constants for stability. Finally, the global Masked SSIM score is obtained by averaging the local map $S$ over the valid mask region:

$$\text{SSIM}_m(\hat{c}_I, \hat{x}_0) = \frac{\sum_p S(p) \cdot m_p}{\sum_p m_p}. \tag{18}$$

**Overall Reward.** The final pixel-level similarity reward $r_{\text{sim}}$ aggregates these metrics:

$$r_{\text{sim}} = w_s \cdot \text{SSIM}_m(c_I, x_0) + (1 - w_s) \cdot \overline{\text{PSNR}}_m, \tag{19}$$

where $w_s$ is a weight set to $0.5$. The total reward $r$ is defined as:

$$r = \text{op}\left( \lambda_{\text{mllm}} r_{\text{mllm}} + \lambda_{\text{sim}} r_{\text{sim}} \right), \tag{20}$$

where $\text{op}(\cdot)$ denotes the optimality probability transformation (Zheng et al., 2025).

## B.3. Theoretical Analysis of Region-based Regularization

In this section, we provide a rigorous derivation and analysis of the proposed region-based regularizers, $L_{\text{ner}}^+$ and $L_{\text{er}}^-$. We list their definitions, gradient dynamics, and the corresponding physical interpretations step-by-step. Crucially, we explain the rationale behind applying distinct regularization strategies to implicit positive and negative policies, demonstrating how they possess different interactions with the old policy. For reference, the overall objective $\mathcal{L}(\theta)$ in Section 3.2 is formulated as:

$$\mathcal{L}(\theta) = \mathbb{E}_{c,\pi^{\text{old}}(x_0|c),t}\left[r \cdot (\mathcal{L}^+ + \lambda_{\text{ner}}L_{\text{ner}}^+) + (1-r) \cdot (\mathcal{L}^- + \lambda_{\text{er}}L_{\text{er}}^-)\right]. \tag{21}$$

### B.3.1. POSITIVE REGULARIZER ON NON-EDIT REGION ($L_{ner}^+$)

The positive regularizer aims to maintain background consistency and strictly constrain the high-reward samples to the original input image.

**Definition**. The positive regularizer on non-edit region is defined as:

$$L_{\text{ner}}^+ = \max\left(0, \underbrace{\|P_{\text{ner}}(x_\theta^+) - P_{\text{ner}}(c_I)\|_2}_{d_{\text{ner}}} - \tau^+\right), \tag{22}$$

where $P_{\text{ner}}(\cdot)$ extracts the non-edit region features, and $c_I$ is the input latent. Here, $\tau^+$ is a threshold and constrains the similarity, which should ideally be zero. However, the editing masks of 40K training samples may have imprecise edges and they only cover the editing regions, resulting in pixel value changes in $P_{ner}(\cdot)$. Therefore, we empirically set $\tau^+$ to 0.001 as a tolerance term.

**Expansion Analysis**. Substituting the positive policy composition $x_\theta^+ = (1-\beta)x_0^{\text{old}} + \beta x_0$, we expand the non-edit region projection to a linear combination of historical and current states:

$$P_{\text{ner}}(x_\theta^+) = (1-\beta)P_{\text{ner}}(x_0^{\text{old}}) + \beta P_{\text{ner}}(x_0). \tag{23}$$

**Drift Decomposition**. The total background distance $d_{\text{ner}}$ can be decomposed into the weighted sum of the *Anchor Bias* (from the old policy) and the *Current Drift* (from the new model):

$$d_{\text{ner}} = \left\|(1-\beta)\underbrace{[P_{\text{ner}}(x_0^{\text{old}}) - P_{\text{ner}}(c_I)]}_{\Delta_{\text{old}} \text{ (Anchor Bias)}} + \beta\underbrace{[P_{\text{ner}}(x_0) - P_{\text{ner}}(c_I)]}_{\Delta_{\text{new}} \text{ (Current Drift)}}\right\|_2. \tag{24}$$

**Reward Maximization with Anchor Compensation**. To analyze the joint effect of reinforcement guidance and regularization, we combine the optimal solution from Equation (13) with the gradient descent direction of $L_{\text{ner}}^+$. The final velocity update $v_\theta^{\text{new}}$ acts as a superposition of a *driving force* (for editing) and a *corrective force* (for consistency):

$$v_\theta^{\text{new}} \leftarrow \underbrace{v^{\text{old}}(\boldsymbol{x}_t, \boldsymbol{c}, t) + \frac{2}{\beta}\Delta(\boldsymbol{x}_t, \boldsymbol{c}, t)}_{\text{Driving Force}} - \underbrace{\mathbb{I}(d_{\text{ner}} > \tau^+) \cdot \beta t \cdot \mathbf{u}_{\text{ner}}}_{\text{Corrective Force (Regularizer)}}, \tag{25}$$

where $\mathbb{I}(\cdot)$ is the indicator function, and the unit drift direction is $\mathbf{u}_{\text{ner}} = \frac{(1-\beta)\Delta_{\text{old}} + \beta\Delta_{\text{new}}}{d_{\text{ner}}}$. This formulation reveals a critical compensation mechanism. (1) Driving Force: The term $\frac{2}{\beta}\Delta$ represents the gradient direction that maximizes the semantic alignment reward, pushing the generated sample towards the target edit. (2) Anchor Compensation: When the background drift is significant ($d_{\text{ner}} > \tau^+$), the corrective force activates. To minimize the total drift vector, the new model's deviation $\Delta_{\text{new}}$ is forced to point in the *opposite* direction of the historical anchor bias $\Delta_{\text{old}}$. Consequently, the velocity field is optimized to actively "pull" the generated image back towards $c_I$, counteracting the accumulated error from the old policy while pursuing the editing goal.

**Role of Hinge Loss**. The threshold $\tau^+$ creates a tolerance zone. Small, imperceptible background variations are ignored to preserve generation diversity. The penalty only activates when the combined structural drift $(1-\beta)\Delta_{\text{old}} + \beta\Delta_{\text{new}}$ becomes significant, preventing the model from collapsing into a trivial identity mapping while ensuring background fidelity.

B.3.2. NEGATIVE REGULARIZER ON EDIT REGION ($L_{er}^-$)

The negative regularizer forces the model to perform meaningful edits in low-reward (under-edited) samples.

**Definition**. The negative regularizer on edit region is defined as:

$$L_{er}^- = \max\left(0, \tau^- - \underbrace{\|P_{er}(x_\theta^-) - P_{er}(c_I)\|_2}_{d_{er}}\right). \tag{26}$$

It penalizes the model when the edit region of the negative sample is too similar to the input. Here, the threshold $\tau^-$ cannot be set as a constant due to the differences among various editing types. For instance, removing the whole object causes larger pixel changes than editing the action. Thus, we directly use the distance between the latent of the input image $c_I$ and the latent of the corresponding edited sample $x_0$, $\tau^- = \|P_{er}(x_0) - P_{er}(c_I)\|_2$, as the threshold, achieving an adaptive thresholding strategy.

**Expansion Analysis**. With the negative policy composition $x_\theta^- = (1+\beta)x_0^{old} - \beta x_0$, we have:

$$P_{er}(x_\theta^-) = (1+\beta)P_{er}(x_0^{old}) - \beta P_{er}(x_0). \tag{27}$$

Note that the negative coefficient for $x_0$ characterizes the negative policy update direction.

**Distance Expression**. It can be derived that:

$$d_{er} = \left\|(1+\beta)\underbrace{[P_{er}(x_0^{old}) - P_{er}(c_I)]}_{\Gamma_{old}} - \beta\underbrace{[P_{er}(x_0) - P_{er}(c_I)]}_{\Gamma_{new}}\right\|_2. \tag{28}$$

**Reward Maximization with Edit Amplification**. Similar to the positive regularizer, we analyze the joint effect by combining the optimal policy update with the gradient descent direction of $L_{er}^-$. The final velocity update $v_\theta^{new}$ acts as a superposition of the driving force and an *expansive force* that prevents laziness:

$$v_\theta^{new} \leftarrow \underbrace{v^{old}(\boldsymbol{x}_t, \boldsymbol{c}, t) + \frac{2}{\beta}\Delta(\boldsymbol{x}_t, \boldsymbol{c}, t)}_{\text{Driving Force}} - \underbrace{\mathbb{I}(d_{er} < \tau^-) \cdot \beta t \cdot \mathbf{u}_{er}}_{\text{Expansive Force (Regularizer)}}, \tag{29}$$

where the unit direction of the edit deviation is $\mathbf{u}_{er} = \frac{(1+\beta)\Gamma_{old} - \beta\Gamma_{new}}{d_{er}}$. This formulation highlights the repulsive mechanism of the negative regularizer. (1) Expansive Force: When the edit magnitude is insufficient ($d_{er} < \tau^-$), the regularizer activates. Unlike the positive case where the force pulls the model back, here the gradient term $-\beta t \cdot \mathbf{u}_{er}$ interacts with the negative coefficient ($-\beta$) of $x_0$ in the composite policy. (2) Mechanism: By subtracting a vector aligned with $\mathbf{u}_{er}$, the optimization effectively increases the magnitude of the difference vector $(1+\beta)\Gamma_{old} - \beta\Gamma_{new}$. This "pushes" the generated content away from the input $c_I$, forcing the model to amplify the editing effect and break away from the initial state.

**Physical Meaning**. Despite the mathematical similarity to the positive gradient, the physics are inverted by the condition $d_{er} < \tau^-$. The loss forces the distance to *increase*. Since $x_0$ contributes negatively to $x_\theta^-$, the gradient updates $x_0$ to amplify the difference between the composite output and the input $c_I$, effectively "pushing" the model out of the lazy regime.

**Role of Hinge Loss**. The hinge acts as a "minimum effort" constraint. Once the edit is sufficiently distinct from the input ($d_{er} \geq \tau^-$), the gradient vanishes. This encourages diversity without imposing an upper bound on the magnitude of change.

**Interaction with Old Policy**. We provide an explanation of why $L_{er}^-$ does not induce random noise or meaningless structure:

- **Relative Update:** The gradient depends on $\Gamma_{old}$. If the old policy fails to edit ($\Gamma_{old} \approx 0$), the regularization forces $x_0$ to generate a change. The update is relative to the old policy's state, not a random perturbation.

- **Reward Constraint:** While $L_{er}^-$ provides the *magnitude* of the change (pushing away from $c_I$), the preference optimization loss provides the *direction* (aligning with the text prompt). The model cannot simply generate noise to satisfy $L_{er}^-$, as that would violate the reward objective. The optimal solution is to generate a meaningful structure that is both distinct from the input (satisfying regularization) and semantically correct (satisfying reward).

### B.3.3. SUMMARY: ASYMMETRIC DESIGN RATIONALE

The separation of regularizers for positive and negative policies is grounded in their distinct roles during optimization:

1. Positive policy represents the "safe" trajectory. The risk here is background corruption. Thus, $L_{\text{ner}}^+$ acts as an **anchor**, using the old policy's history to strictly bound the non-edit region to the input.

2. Negative policy represents the "rejected" trajectory, often characterized by under-editing (laziness). Thus, $L_{\text{er}}^-$ acts as a **catalyst**. It leverages the old policy's failure (lack of change) to force a structural deviation.

By conditioning these regularizers on the old policy, we ensure that the updates are history-aware corrections rather than arbitrary perturbations, balancing structural stability with editing capability.

### B.4. The Pseudo-code for CoCoEdit Training

---

**Algorithm 1** CoCoEdit Training

---

**Require:** Pretrained diffusion policy $v^{\text{ref}}$, prompt dataset $\{c\}$, input image latent $c_I$, non-edit mask $m$.
 1: **Initialize:** Data collection policy $v^{\text{old}} \leftarrow v^{\text{ref}}$, training policy $v_\theta \leftarrow v^{\text{ref}}$, data buffer $\mathcal{D} \leftarrow \emptyset$.
 2: **for** each iteration $i$ **do**
 3:     **for** each sampled prompt $c$ **do**
 4:         Collect $K$ clean images $x_0^{1:K}$ using $v^{\text{old}}$.
 5:         **Reward Collection:**
 6:             Calculate MLLM reward: $r_{\text{mllm}} = \text{avg}(\text{Editing Effect, Visual Quality})$.
 7:             Calculate Similarity reward: $r_{\text{sim}} = \text{avg}(\text{PSNR, SSIM})$ between $x_0$ and $c_I$ masked by $m$.
 8:             Aggregate raw rewards: $r^{\text{raw}} = \lambda_{\text{mllm}} r_{\text{mllm}} + \lambda_{\text{sim}} r_{\text{sim}}$.
 9:         Normalize raw rewards: $r^{\text{norm}} := r^{\text{raw}} - \text{mean}(\{r^{\text{raw}}\}^{1:K})$.
10:         Define optimality probability $r = 0.5 + 0.5 * \text{clip}\{r^{\text{norm}}/Z_c, -1, 1\}$.
11:         $\mathcal{D} \leftarrow \{c, x_0^{1:K}, r^{1:K} \in [0, 1]\}$.
12:     **end for**
13:     **for** each mini batch $\{c, x_0, r\} \in \mathcal{D}$ **do**
14:         Forward diffusion process: $x_t = \alpha_t x_0 + \sigma_t \epsilon$; $v = \dot{\alpha}_t x_0 + \dot{\sigma}_t \epsilon$.
15:         Implicit positive velocity: $v_\theta^+(x_t, c, t) := (1 - \beta)v^{\text{old}}(x_t, c, t) + \beta v_\theta(x_t, c, t)$.
16:         Implicit negative velocity: $v_\theta^-(x_t, c, t) := (1 + \beta)v^{\text{old}}(x_t, c, t) - \beta v_\theta(x_t, c, t)$.
17:         **Compute Diffusion Loss:** Predict $\hat{x}_0^+$ from $v_\theta^+$ and $\hat{x}_0^-$ from $v_\theta^-$.
            $\mathcal{L}^+ = \|x_\theta^+(x_t, c, t) - x_0\|_2^2, \mathcal{L}^- = \|x_\theta^-(x_t, c, t) - x_0\|_2^2$
18:         **Compute Region-based Regularizers:**
19:         $L_{\text{ner}}^+ = \max(0, d(x_\theta^+(x_t|c), c_I)_{\tilde{m}} - \tau^+), L_{\text{er}}^- = \max(0, \tau^- - d(x_\theta^-(x_t|c), c_I)_{1-\tilde{m}})$
20:         **Update Parameters:**
21:         $\mathcal{L}_{\text{total}} = r \cdot (\mathcal{L}^+ + \lambda_{\text{ner}} L_{\text{ner}}^+) + (1 - r) \cdot (\mathcal{L}^- + \lambda_{\text{er}} L_{\text{er}}^-)$
22:         $\theta \leftarrow \theta - \lambda \nabla_\theta \mathcal{L}_{\text{total}}$
23:     **end for**
24:     Update data collection policy $\theta^{\text{old}} \leftarrow \eta_i \theta^{\text{old}} + (1 - \eta_i)\theta$, and clear buffer $\mathcal{D} \leftarrow \emptyset$.
25: **end for**
26: **Output:** $v_\theta$

---

## C. More Experimental Details and Results

### C.1. Implementation Details

Following the training setup in (Lin et al., 2025), we train all models with learning rate set to $3e^{-4}$ at a resolution of $512 \times 512$. During sampling, we set sampling inference steps to 10, the number of images per prompt to 12, and the number of groups to 24; for training, we set the KL loss weight to 0.0001. Regarding model-specific hyperparameters, for FLUX.1 Kontext, we set the guidance strength to 2.5, and both the positive background regularization weight ($\lambda_{\text{ner}}$) and negative edit

regularization weight ($\lambda_{er}$) to 0.2. For Qwen-Image-Edit, we set the guidance strength to 1.0, with $\lambda_{ner}$ set to 0.5 and $\lambda_{er}$ set to 0.2. The entire training process takes approximately 4 days on 8 NVIDIA A800 GPUs.

During the inference stage, we follow the default settings of each model. For BAGEL (Deng et al., 2025), we set the number of inference steps to 50, the text guidance scale to 4.0, and the image guidance scale to 2.0. For OmniGen2 (Wu et al., 2025b), we set the number of inference steps to 50, text guidance scale to 5.0, and image guidance scale to 2.0. For Step1X-Edit (Liu et al., 2025b), we set the number of inference steps to 28 and the true guidance scale to 4.0. For the variants of FLUX.1 Kontext, including "w/ Edit R1" and "w/ CoCoEdit", we set the number of inference steps to 28 and the guidance scale to 2.5. For the variants of Qwen-Image-Edit, including "w/ Edit R1", "w/ MotionNFT" and "w/ CoCoEdit", we set the number of inference steps to 40, the true guidance scale to 4.0, and the guidance scale to 1.0.

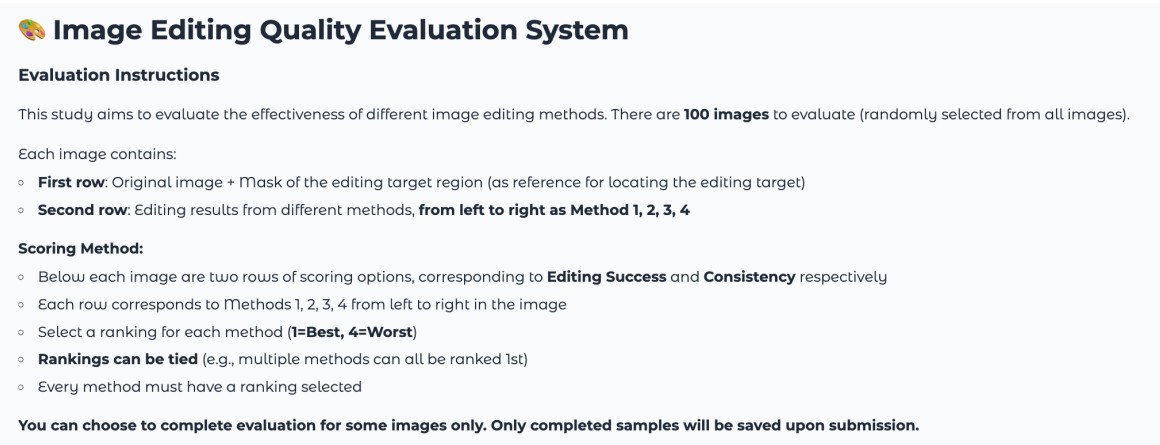

*Figure 9.* The description of display layout and the evaluation standard in the web-based evaluation system.

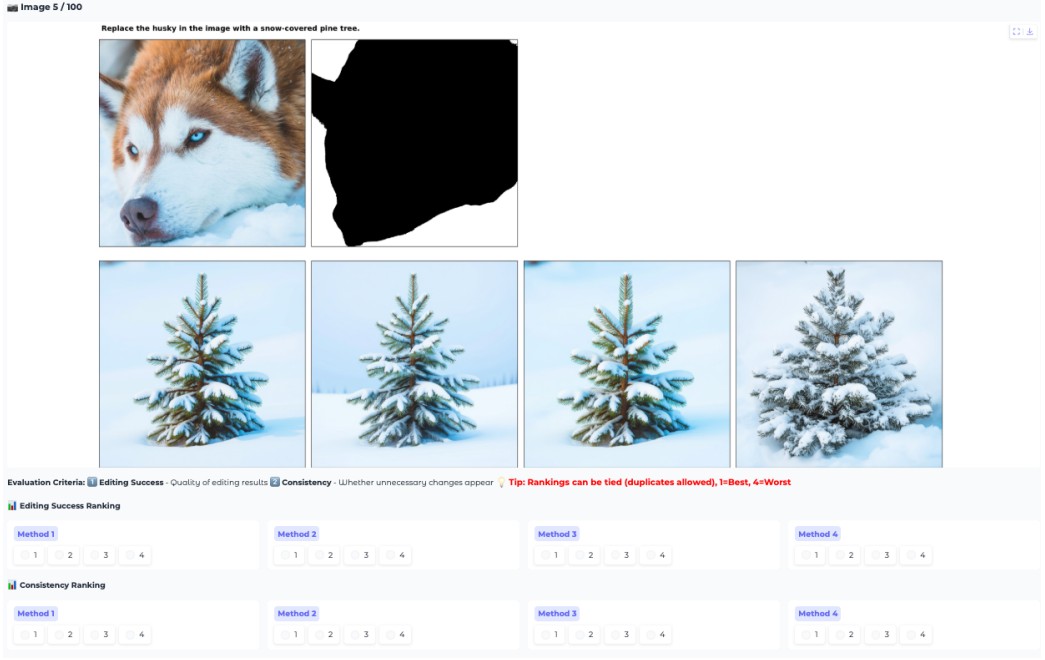

*Figure 10.* One sample of the user study. The users are required to rank the editing results of the input image.

## C.2. Details of User Study

As shown in Figures 9 and 10, we developed a web-based evaluation system using Gradio to collect human ratings on image editing quality. The system contains 100 randomly sampled images (with seed 42 for reproducibility, the snapshots of these images are shown in Figure 11) from our evaluation dataset, and their editing results by the variants of Qwen-Image-Edit and FLUX.1 Kontext models.

We invite 30 volunteers, most of whom are researchers in computer vision, to participate in the evaluation. As shown in Figure 10, for each image to be edited, the evaluators are presented a composite visualization containing the original image, the mask indicating the target editing region, and the editing results from multiple methods (FLUX.1 Kontext has 3 variants and Qwen-Image-Edit has 4 variants, as shown in Tables 1 and 2). For each sample, evaluators are instructed to provide rankings based on two key dimensions: (1) **Editing Quality**, *i.e.*, whether the image is successfully edited according to the instruction, and (2) **Consistency Preservation**, *i.e.*, whether unnecessary changes are introduced to non-target regions. The ranking system allows for ties (*e.g.*, multiple methods can be ranked first), with ratings ranging from 1 (best) to 3 (for FLUX.1 Kontext) or 4 (for Qwen-Image-Edit) (worst), depending on the number of competing models. Each submission generates a timestamped JSON file containing the user ID, rankings, and metadata, preventing duplicate submissions through session state management.

Finally, we collect all the results for the groups of FLUX.1 Kontext and Qwen-Image-Edit, respectively. The average rank of each variant of FLUX.1 Kontext and Qwen-Image-Edit is calculated. Results are shown in Tables 1 and 2 of the main paper.

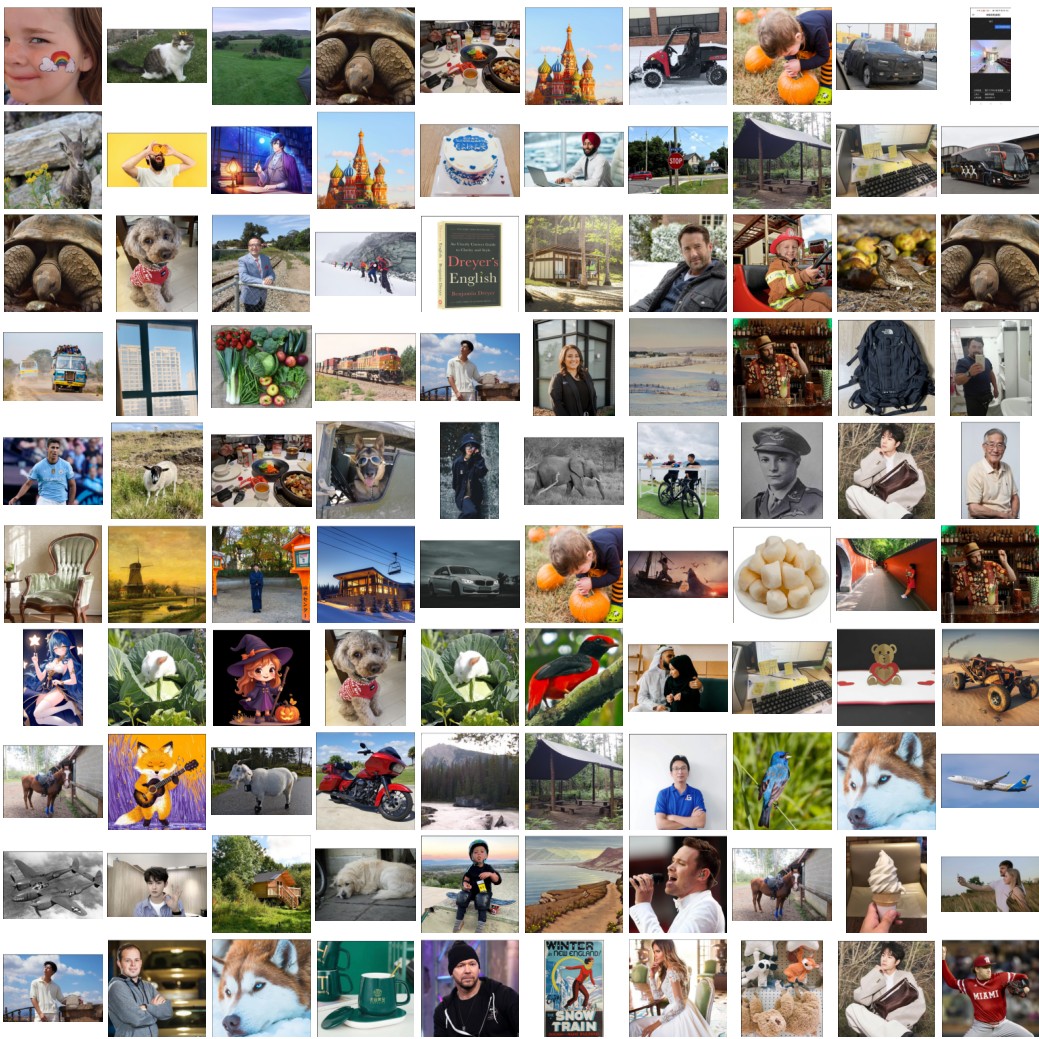

*Figure 11.* The snapshots of all the samples used in user study.

| Model | Background | Color | Material | Motion | Ps_human | Add | Remove | Replace | Text | Overall ↑ | PSNR | SSIM | Rank ↓ |
|---|---|---|---|---|---|---|---|---|---|---|---|---|---|
| GPT-Image-1 | **7.934** | **7.944** | **7.548** | 7.750 | **7.605** | **7.958** | 7.958 | 7.836 | 8.213 | **7.861** | 12.285 | 0.406 | - |
| Nano-Banana | 7.860 | 7.388 | 5.223 | 7.411 | 6.323 | 7.706 | 6.570 | 7.715 | 7.362 | 7.062 | 19.540 | 0.640 | - |
| FLUX.1 Kontext | 7.153 | 6.908 | 5.764 | 5.196 | 5.330 | 7.167 | 6.728 | 6.717 | 5.612 | 6.286 | 24.168 | 0.825 | 2.1 |
| w/ CoCoEdit (Ours) | 7.144 | 7.228 | 6.647 | 6.615 | 6.010 | 7.784 | 7.252 | 7.001 | 6.772 | 6.939 | **25.331** | **0.874** | **1.6** |
| Qwen-Image-Edit | 7.894 | 7.588 | 6.990 | 7.359 | 6.776 | 7.836 | 7.783 | 7.865 | 7.945 | 7.560 | 19.488 | 0.662 | 2.7 |
| w/ CoCoEdit (Ours) | 7.898 | 7.675 | 7.329 | **7.758** | 7.001 | 7.931 | **7.975** | 7.876 | 8.331 | 7.754 | 22.283 | 0.774 | **1.4** |

*Table 5.* Quantitative comparison with commercial models on GEdit-Bench-EN. The best and second-best results are highlighted in **bold** and underlined, respectively.

| Model | Add | Adjust | Replace | Remove | Background | Compose | Action | Overall ↑ | PSNR | SSIM | Rank ↓ |
|---|---|---|---|---|---|---|---|---|---|---|---|
| GPT-Image-1 | 3.81 | 3.86 | **3.78** | 4.32 | **3.85** | **3.10** | 3.87 | **3.80** | 12.209 | 0.308 | - |
| Nano-Banana | 3.79 | 3.90 | 3.68 | 4.42 | 3.75 | 2.97 | 3.82 | 3.76 | 16.926 | 0.449 | - |
| FLUX.1 Kontext | 3.53 | 3.60 | 3.60 | 3.31 | 3.36 | 2.80 | 3.78 | 3.43 | 19.591 | 0.592 | 2.2 |
| w/ CoCoEdit (Ours) | 3.64 | 3.53 | 3.57 | 3.85 | 3.33 | 2.82 | **3.87** | 3.52 | **20.747** | **0.635** | **1.5** |
| Qwen-Image-Edit | 3.80 | 3.77 | 3.66 | 4.33 | 3.63 | 2.92 | 3.78 | 3.70 | 17.635 | 0.526 | 2.7 |
| w/ CoCoEdit (Ours) | **3.85** | **3.96** | 3.76 | **4.47** | 3.74 | 2.87 | 3.85 | 3.79 | 19.125 | 0.555 | **1.7** |

*Table 6.* Quantitative comparison with commercial models on ImgEdit-Bench. The best and second-best results are highlighted in **bold** and underlined, respectively.

## C.3. Comparison with Commercial Models

In Tables 5 and 6, we provide the quantitative comparisons between CoCoEdit and commercial models, including GPT-Image-1 (OpenAI, 2025) and Nano-Banana (Google, 2025). One can see that GPT-Image-1 achieves the best MLLM scores in most of the editing types, but shows severe degradation in pixel-level similarity metrics (about 12dB of PSNR on both benchmarks), causing poor content consistency. Nano-Banana compromises the MLLM scores and presents better similarity metrics than GPT-Image-1. However, its capability of content consistency preservation remains limited. Compared to the commercial models, our CoCoEdit models achieve comparable performance on MLLM scores and significantly superior pixel-level consistency. For example, Qwen-Image-Edit with CoCoEdit provides either the best or the second best scores on GEdit-Bench-EN. On ImgEdit-Bench, Qwen-Image-Edit with our CoCoEdit achieves the second best overall score, only 0.01 lower than GPT-Image-1. In the comparison of pixel-level similarity metrics, CoCoEdit exhibits conspicuous advantages than both the open-source baseline models and the commercial models, demonstrating the effectiveness of our method in improving the capability of content-consistent editing.

## C.4. Comparison with In-Context and Mask-guided Models

As shown in Table 7, FLUX.1-Fill-dev (Labs, 2024) achieves good consistency scores due to the mask guidance. However, it has low overall editing scores (e.g., 3.258 on GEdit). ICEdit (Zhang et al., 2025) achieves better editing effects, but its consistency is poor.

| Model | GEdit-Bench | | | ImgEdit-Bench | | |
|---|---|---|---|---|---|---|
| | Overall | PSNR | SSIM | Overall | PSNR | SSIM |
| FLUX.1-Fill-dev | 3.258 | **26.793** | 0.786 | 2.04 | **26.463** | **0.701** |
| ICEdit | 5.898 | 21.698 | 0.730 | 2.88 | 21.561 | 0.614 |
| FLUX.1 Kontext w/ CoCoEdit | 6.939 | 25.331 | **0.874** | 3.52 | 20.747 | 0.635 |
| Qwen-Image-Edit w/ CoCoEdit | **7.754** | 22.283 | 0.774 | **3.79** | 19.125 | 0.555 |

*Table 7.* Quantitative comparison with in-context and mask-guided models on GEdit-Bench and ImgEdit-Bench. The best results are highlighted in **bold**.

## C.5. Analysis on the Correlation between MLLM Scores and Pixel-level Similarity Metrics

While MLLMs (Bai et al., 2025; Luo et al., 2025) have demonstrated impressive capabilities in evaluating semantic alignment and aesthetic quality, their ability to assess fine-grained content preservation remains limited. To investigate this, we conducted a quantitative analysis on 100K randomly sampled image pairs from the ImgEdit (Ye et al., 2025) dataset. Each pair consists of an original and an edited image. We calculate PSNR and SSIM values within the non-edit regions.

Meanwhile, we prompt Qwen2.5-VL-32B to evaluate the consistency of non-edit regions. To ensure focusing on background preservation, we designed a rigorous prompt emphasizing pixel-level fidelity, as shown in Figure 12.

As illustrated in Figure 13, the results reveal a significant discrepancy between MLLM and pixel-level metrics. While PSNR and SSIM exhibit a strong positive correlation (Pearson $r = 0.695$), indicating a consistent measurement of signal quality, the MLLM scores show negligible correlation with both PSNR (Pearson $r = 0.039$) and SSIM (Pearson $r = -0.024$). The scatter plots demonstrate that even for image pairs with very low PSNR/SSIM values—indicating severe pixel-level degradation—the MLLM still assigns high scores (*e.g.*, 9 or 10). Conversely, samples with high PSNR and SSIM values may not receive high scores from the MLLM. These findings suggest that current MLLMs are not able to detect fine-grained discrepancies but tend to prioritize high-level semantic consistency. Consequently, relying solely on MLLM-based evaluation is insufficient for content-consistent image editing, demonstrating the necessity of our hybrid evaluation protocol that employs PSNR/SSIM to implement MLLM.

---

**System Prompt for Consistency Evaluation**

**Role:** You are an expert in image quality assessment specializing in detecting fine-grained visual artifacts.

**Task:** You will be provided with three inputs: an original image, an edited image, and a binary mask indicating the edited region. Your goal is to evaluate the consistency of the unedited regions (background) between the two images.

**Step 1:** Analyze the binary mask. The white area represents the edited object, while the black area represents the background that should remain unchanged.
**Step 2:** Compare the background regions of the Original Image and the Edited Image pixel by pixel. Look for unintended changes such as color shifts, bleeding, blurring, noise, or structural distortions.
**Step 3:** Assign a consistency score based on the severity of the artifacts found in the background.

**Requirements:** (1) Ignore the content within the masked edited region entirely; focus ONLY on the background. (2) Be strictly critical of pixel-level fidelity. (3) A score of 10 indicates a pixel-perfect match, while a score of 1 indicates severe background degradation.

**Output Format:** Output the result in the following strict JSON format:

```
{"reasoning": "<brief analysis of background changes>",
  "consistency_score": <int, 1-10>}
```

Make sure the output is strictly valid JSON, with no explanatory text outside the brackets. **Inputs:** Original Image: `<IMAGE_0>`, Edited Image: `<IMAGE_1>`, Object Mask: `<IMAGE_2>`

---

*Figure 12.* The structured system prompt used for evaluating background consistency. We explicitly provide the editing mask to help the MLLM distinguish between the edited object and the background, ensuring the evaluation focuses strictly on unedited regions.

## C.6. Ablation Study on $r_{sim}$, $L_{ner}^+$ and $L_{er}^-$

To validate the effectiveness of the major components of our method, we conduct an ablation study on the pixel-level similarity reward $r_{sim}$, the positive regularizer $L_{ner}^+$, and the negative regularizer $L_{er}^-$ in Table 8.

**Effect of Similarity Reward.** Here, the ratio of rewards is 0.8:0.2. Comparing the first two rows, $r_{sim}$ yields a noticeable improvement in PSNR across both benchmarks, which indicates that the similarity reward signal effectively guides the model to maintain the structural integrity of the original image during the editing process. Therefore, $r_{sim}$ can only provide limited improvement of consistency. Finally, removing $r_{sim}$ from the full configuration results in a performance degradation in both metrics. Optimal performance is achieved by the combination of all techniques.

**Effect of Positive Regularizer.** In the third row, $L_{ner}^+$ leads to a significant boost of consistency, achieving the highest PSNR scores of 22.623dB and 19.416dB. This confirms that explicitly penalizing deviations in the non-editing regions is crucial for preserving non-edit regions. However, we observe a slight drop in the Overall editing score, suggesting that an overly strict focus on content preservation on non-edit regions will constrain the generative flexibility in the editing region.

**Effect of Negative Regularizer.** The complete method with $L_{er}^-$ achieves the best Overall editing performance (7.754 on GEdit-Bench and 3.79 on ImgEdit-Bench) while maintaining a highly competitive PSNR. This demonstrates that the negative regularization effectively complements the positive term by encouraging edits, resulting in an optimal balance between editing effects and background consistency.

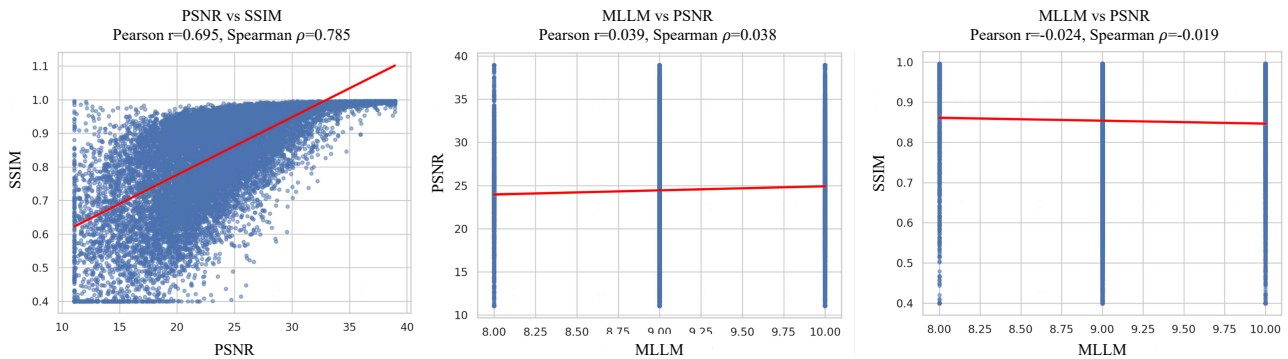

*Figure 13.* Correlation analysis between different consistency evaluation metrics. **Left**: The scatter plot shows a strong positive correlation between traditional metrics PSNR and SSIM (Pearson $r = 0.695$), indicating they capture similar pixel-level characteristics. **Middle & Right**: In contrast, the correlation between the MLLM score and traditional metrics is negligible (Pearson $r \approx 0$). The regression lines are nearly flat. This demonstrates that the MLLM cannot evaluate pixel-level consistency.

The first row of Table 8 (the "baseline") shows the results of Qwen-Image-Edit trained on our proposed CoCoEdit-40K dataset using only the MLLM reward (the same reward used in Edit-R1). This ablation is specifically designed to demonstrate that simply constructing a dataset to apply RL training is not enough to solve the consistency problem. As shown in Table 8, when trained only with the MLLM reward, the model suffers a severe drop in content consistency (e.g., PSNR shows 18.236 dB on GEdit-Bench, which is lower than all baseline methods in Table 1). By introducing our proposed pixel-level similarity reward and region-based regularizers (full CoCoEdit) to utilize our CoCoEdit-40K dataset, the PSNR jumps significantly to 22.283 dB (ranking the 3rd among all the 10 methods in Table 1) and 19.125 dB on ImgEdit (ranking the 4th in Table 2). This shows that the observed improvements in consistency mainly come from our proposed RL framework, rather than the dataset or base model.

| $r_{sim}$ | $L_{ner}^{+}$ | $L_{er}^{-}$ | GEdit-Bench | | ImgEdit-Bench | |
|:---:|:---:|:---:|:---:|:---:|:---:|:---:|
| | | | **Overall** | **PSNR** | **Overall** | **PSNR** |
| | | | 7.739 | 18.236 | 3.73 | 16.437 |
| ✓ | | | 7.731 | 20.152 | 3.73 | 17.832 |
| ✓ | ✓ | | 7.693 | **22.623** | 3.69 | **19.416** |
| ✓ | ✓ | ✓ | **7.754** | 22.283 | **3.79** | 19.125 |
| | ✓ | ✓ | 7.734 | 21.124 | 3.78 | 18.107 |

*Table 8.* Ablation study on $r_{sim}$, $L_{ner}^{+}$, and $L_{er}^{-}$ with Qwen-Image-Edit baseline model.

## C.7. Ablation Study on Reward Ratio

In Section 4.4 of the main paper, we discussed the differences between "ratio=0.5:0.5" and "ratio=0.8:0.2" and the severe reduction of PSNR caused by high ratio of pixel-level similarity reward. In this section, we provide more settings of different ratios without region-based regularizer for specialized analyses of the training stability. As shown in Figure 14, the training process is stable across ratios ranging from 0.7:0.3 to 0.9:0.1, where the performance improvements on GEdit-Bench are comparable and consistent. Among these configurations, the ratio of 0.8:0.2 achieves the most favorable trade-off, becoming our default setting of Qwen-Image-Edit. Furthermore, the results by using only the MLLM reward (ratio=1.0:0.0) show that although the generative score steadily increases, the PSNR suffers a continuous decline. This negative correlation explains why relying exclusively on MLLM feedback leads to a degradation in content consistency.

## C.8. More Visual Results

We provide more qualitative comparisons between CoCoEdit, Edit-R1 (Li et al., 2025), MotionNFT (Wan et al., 2025), and the baseline models, including FLUX.1 Kontext (Labs et al., 2025) and Qwen-Image-Edit (Wu et al., 2025a). Comparisons among variants of Qwen-Image-Edit are shown in Figures 15 to 18, and comparisons among variants of FLUX.1 Kontext

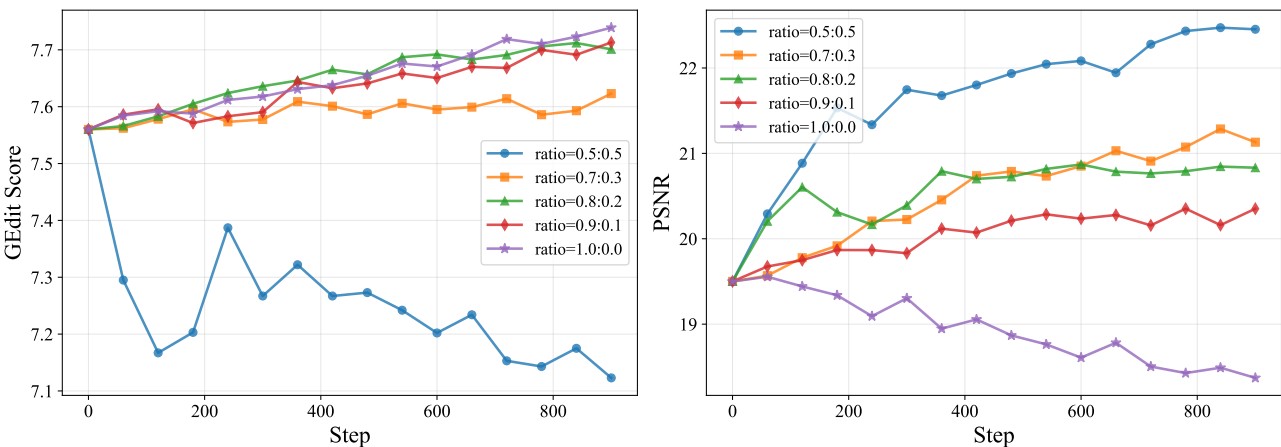

*Figure 14.* Comparison of performance curves of different ratios between $r_{mllm}$ and $r_{sim}$ on Qwen-Image-Edit baseline model.

are shown in Figures 19 to 21. From the visual comparisons, one can see that our CoCoEdit achieves much better content consistency among various editing instructions.

## D. Limitation Analysis and Future Work

Although we have achieved significant advantages in content consistency over previous methods, our CoCoEdit method still has some limitations. First, the quality of our dataset is affected by the visual understanding and reasoning capability of MLLM. Hallucinations in responses can occasionally introduce noise into the training data. Second, physical and perspective inconsistencies remain a challenge due to the intrinsic shortage of baseline models. In Figure 22 (top half), the added objects are not physically plausible. The cat, canoe, and person exhibit incorrect scale ratios or locations. Even for Nano-Banana, the man still looks like walking on the railing (row 1) and the cat is extremely big compared to the sofa (row 3). Thus, current editing models—including ours and baselines—ignore 3D depth awareness in some cases, leading to unnatural fusion in complex perspective scenarios. Third, although CoCoEdit preserves the contents in non-edit regions, in some cases there is a trade-off between attribute editing and texture preservation within the edit region. In Figure 22 (bottom half), the fabric patterns and folds of the clothes or the tile details of the roof are sometimes over-smoothed. Note that texture loss is prevalent even in commercial models, highlighting the difficulty of perfectly disentangling semantic attributes from textural details during editing.

In future work, we will first increase the quality and quantity of our CoCoEdit training dataset with the advancement of MLLM, as well as manual inspection. Second, we will investigate how to improve the physical plausibility of editing results through incorporating 3D priors. Last but not least, how to balance the attribute modification and original property preservation also needs to be investigated.

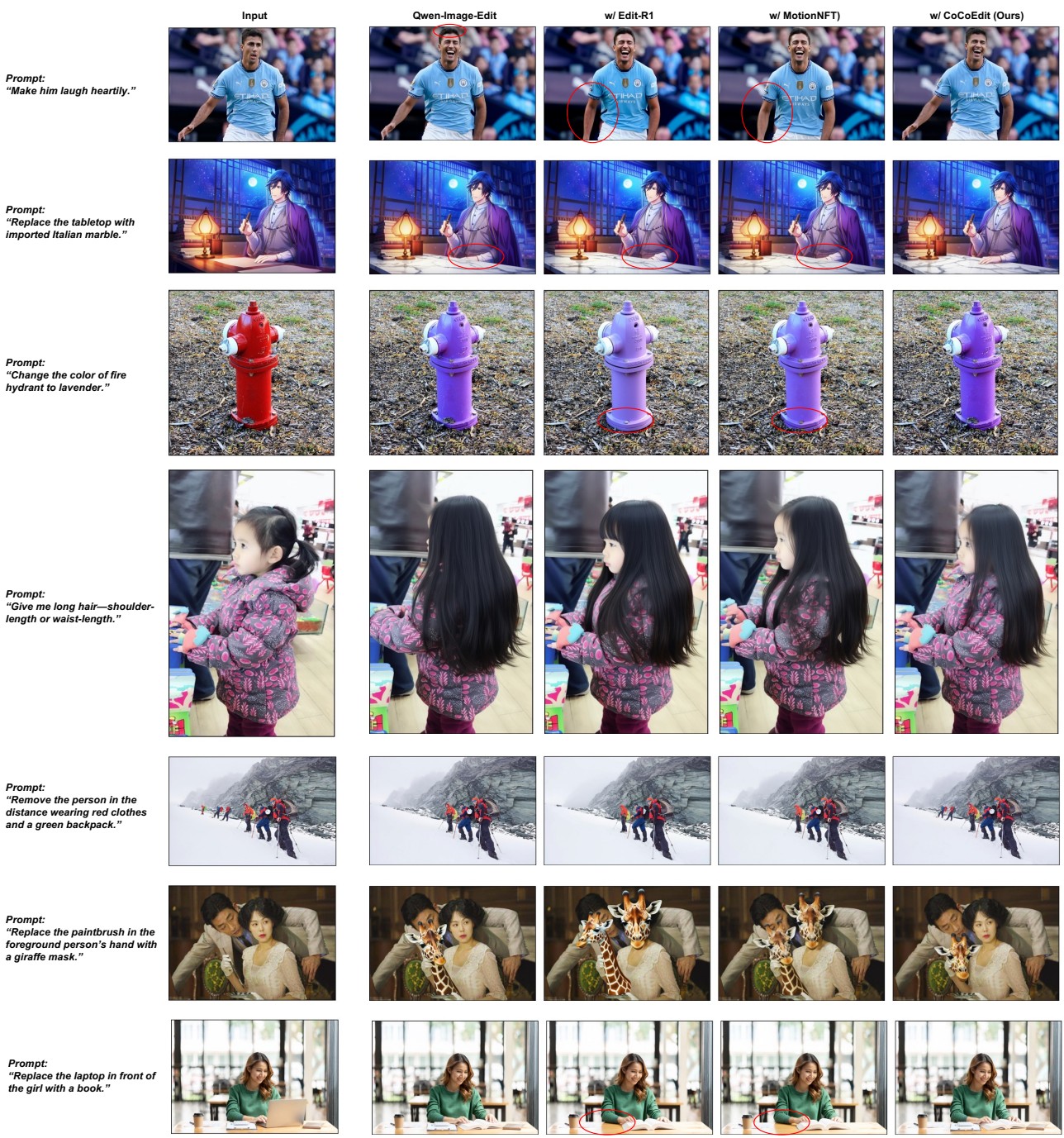

*Figure 15.* Qualitative comparison of Qwen-Image-Edit, "w/ Edit R1", "w/ MotionNFT" and "w/ CoCoEdit".

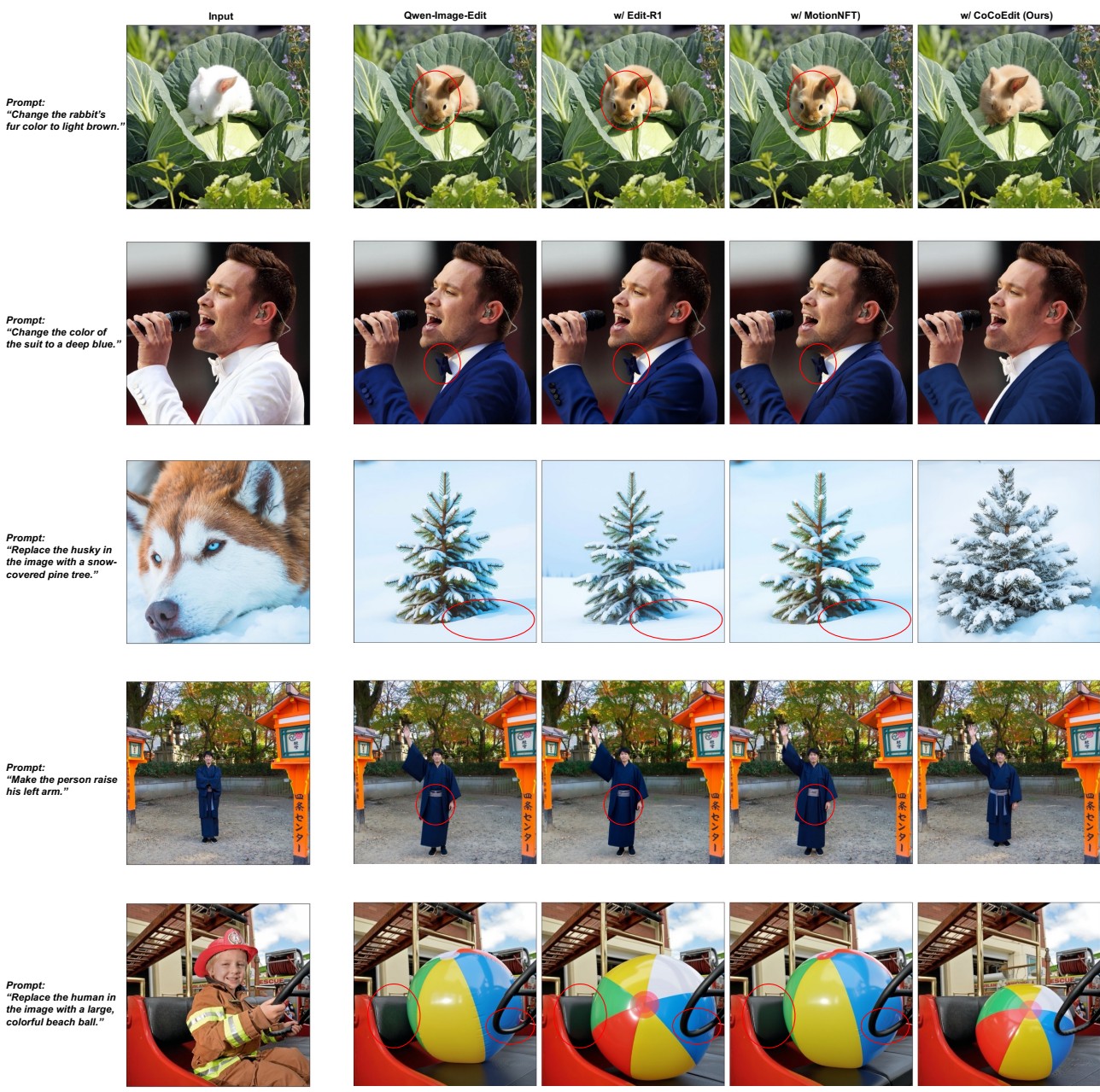

*Figure 16.* Qualitative comparison of Qwen-Image-Edit, "w/ Edit R1", "w/ MotionNFT" and "w/ CoCoEdit".

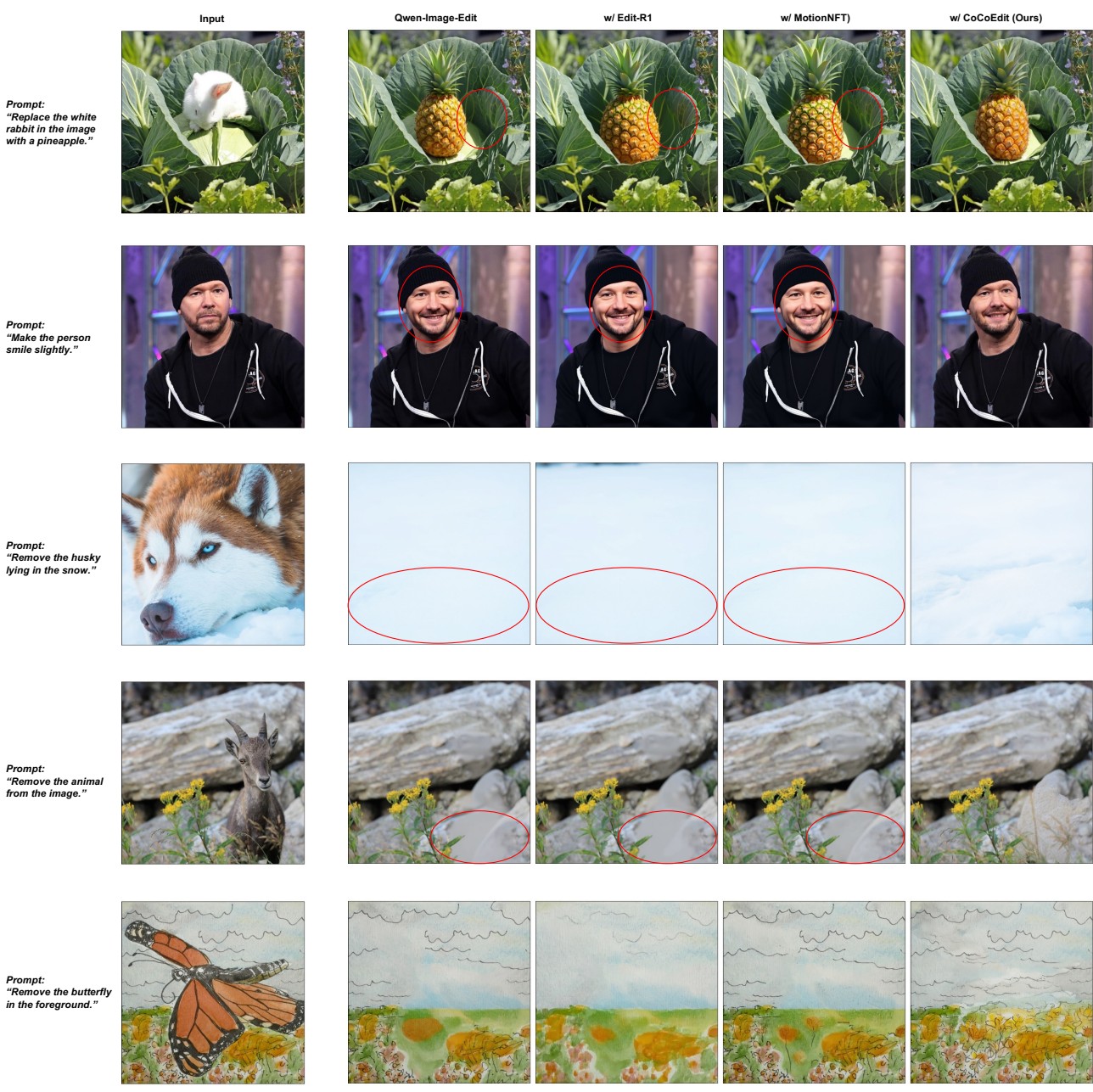

*Figure 17.* Qualitative comparison of Qwen-Image-Edit, "w/ Edit R1", "w/ MotionNFT" and "w/ CoCoEdit".

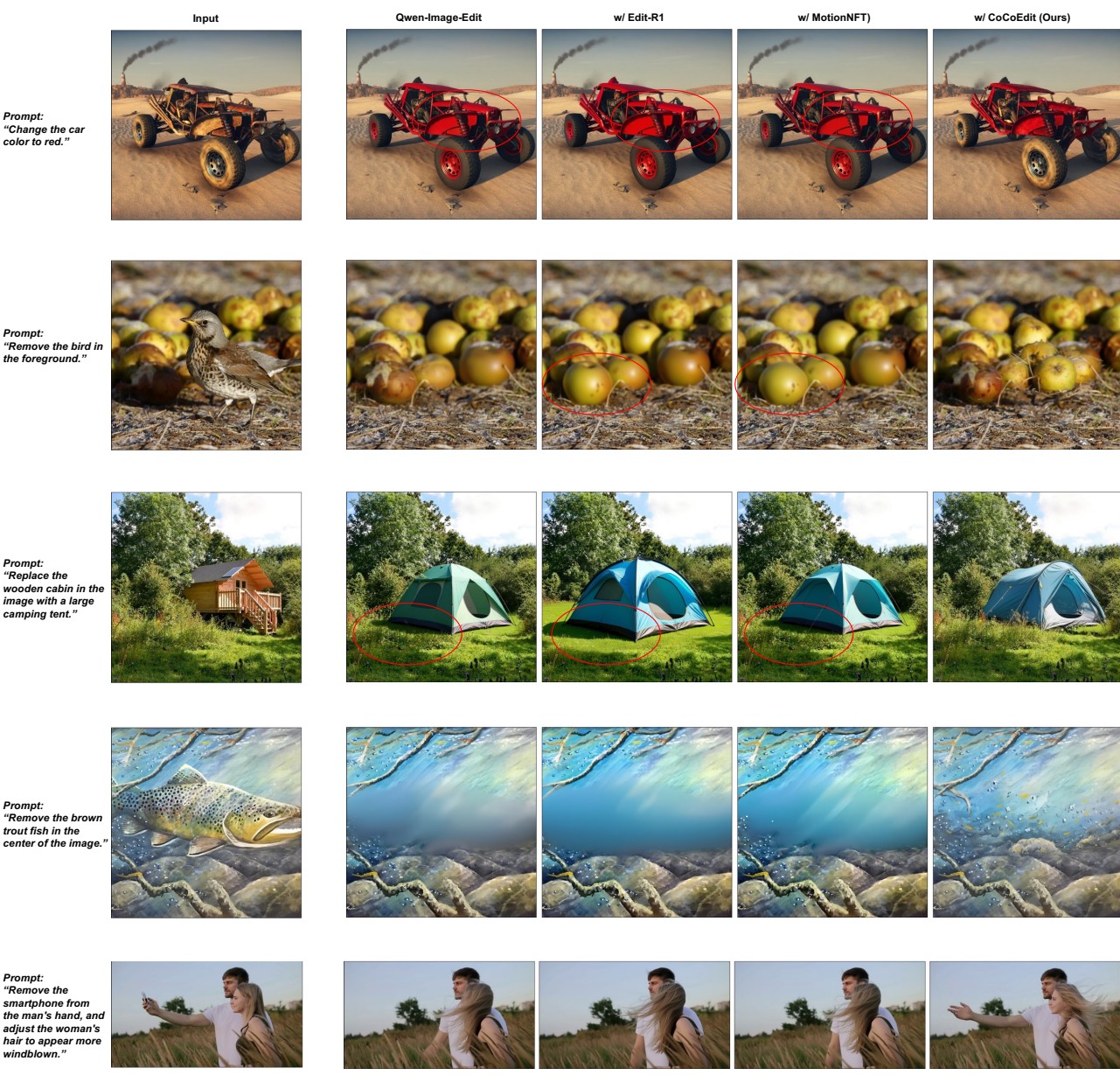

*Figure 18.* Qualitative comparison of Qwen-Image-Edit, "w/ Edit R1", "w/ MotionNFT" and "w/ CoCoEdit".

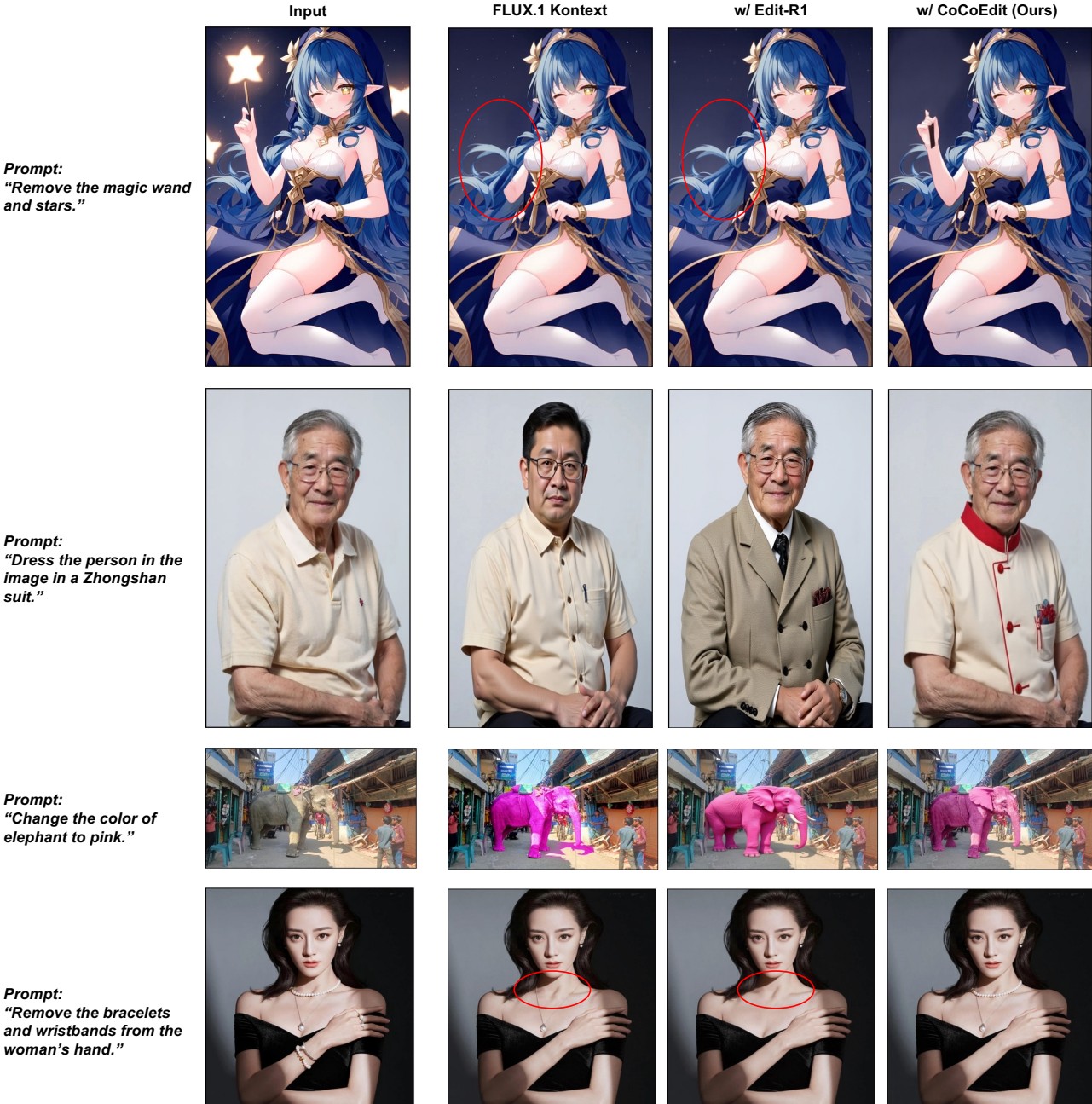

*Figure 19.* Qualitative comparison of FLUX.1 Kontext, "w/ Edit R1", and "w/ CoCoEdit".

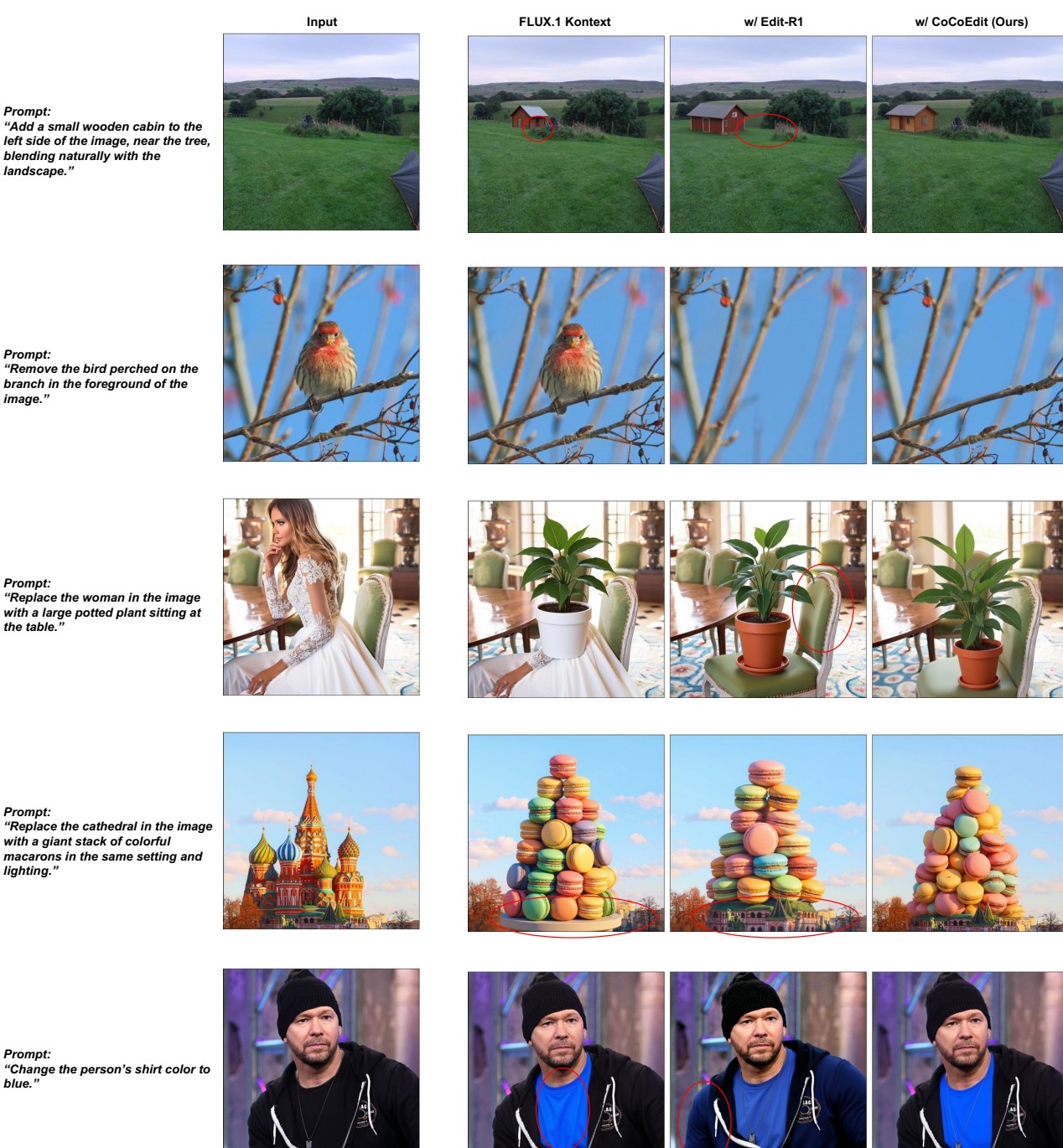

*Figure 20.* Qualitative comparison of FLUX.1 Kontext, "w/ Edit R1", and "w/ CoCoEdit".

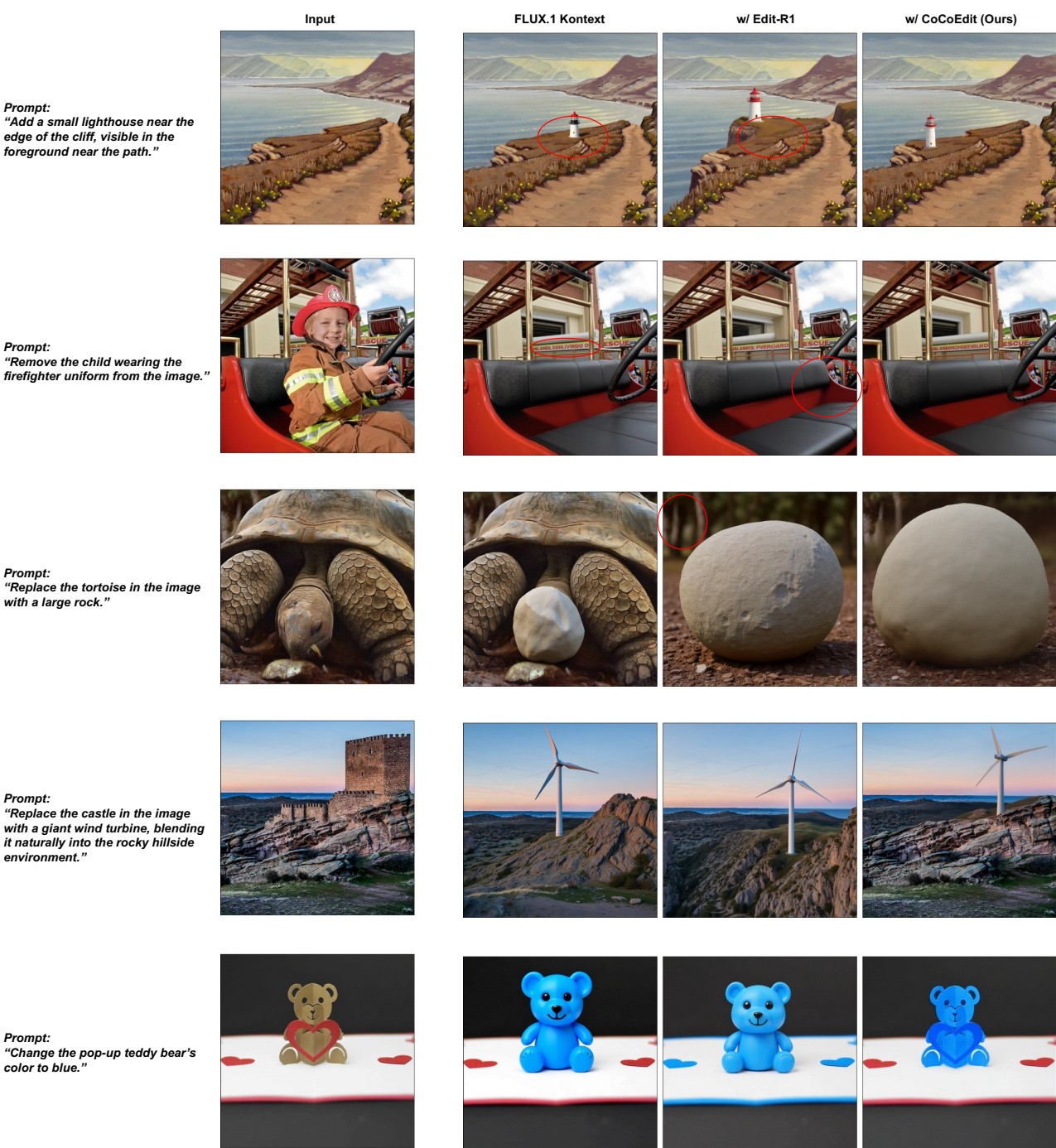

*Figure 21.* Qualitative comparison of FLUX.1 Kontext, "w/ Edit R1", and "w/ CoCoEdit".

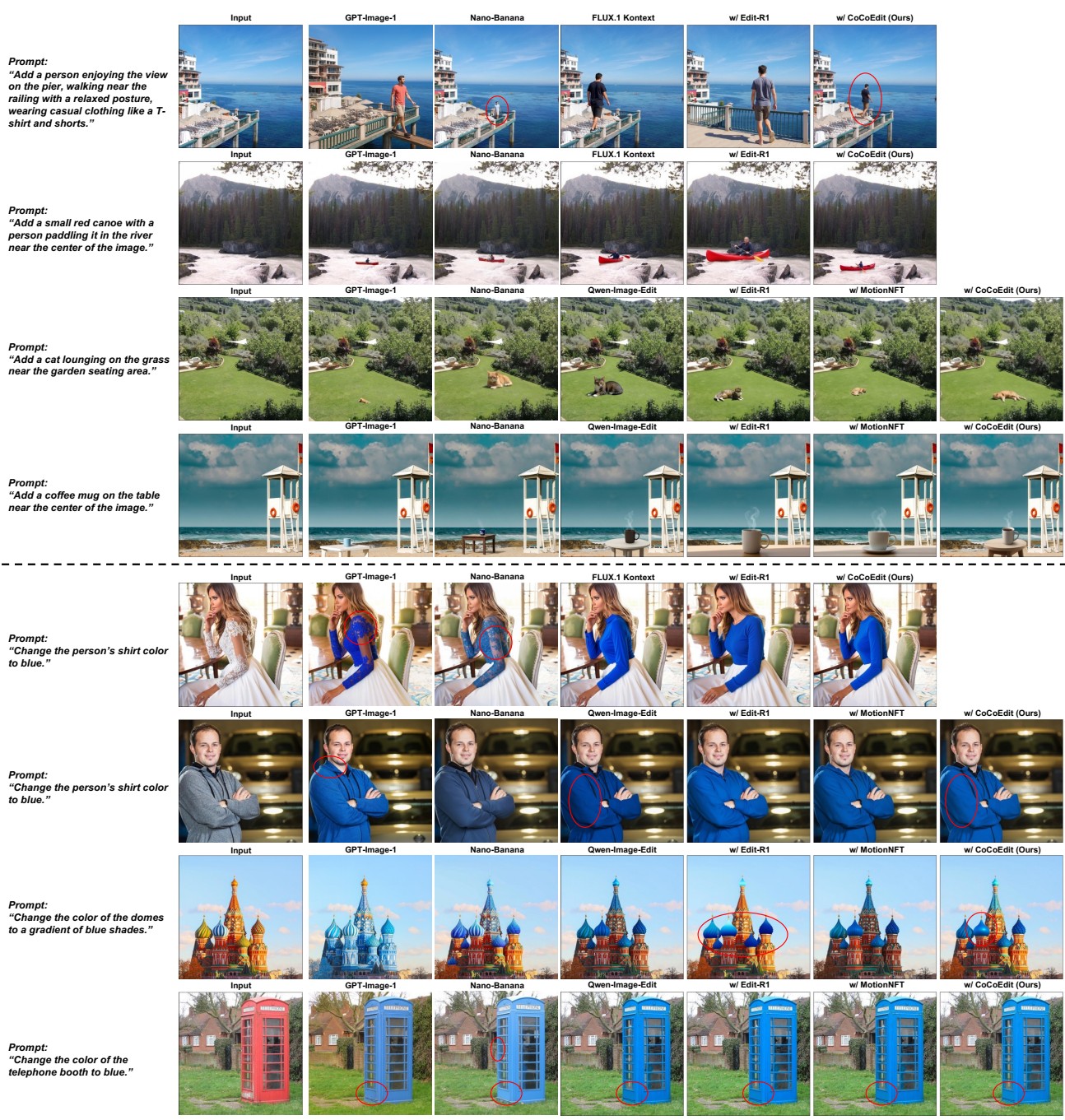

*Figure 22.* Limitations and failure cases. While CoCoEdit achieves robust editing, it shares common challenges with state-of-the-art methods and is limited by the capability of baseline models (Qwen-Image-Edit and FLUX.1 Kontext). Top rows: Issues with physical plausibility (*e.g.*, scale and perspective) when adding objects. Bottom rows: Loss of fine-grained textures within edit region (*e.g.*, patterns and folds) when performing strong color alterations.

