# OpenReview forum: "CoCoEdit: Content-Consistent Image Editing via Region Regularized Reinforcement Learning"
_ICML.cc/2026/Conference — ICML 2026 regular_

### Official Review · Reviewer_UsUV · 2026-03-01

**Soundness:** 3
**Presentation:** 3
**Significance:** 3
**Originality:** 3
**Overall Recommendation:** 4
**Confidence:** 4

**Summary:**

This paper introduces CoCoEdit, a post-training framework for content-consistent image editing via region-regularized reinforcement learning. The key insight is that current large editing models achieve strong editing effects but create unwanted changes in non-edit regions, leading to low content consistency (measured by PSNR/SSIM). The authors propose two main technical contributions: (1) a pixel-level similarity reward (PSNR/SSIM on non-edit regions) to complement MLLM-based rewards, and (2) region-based regularizers that apply different constraints to edit vs non-edit regions. To enable training, they construct CoCoEdit-40K, a dataset of 40K samples with refined instructions and editing masks. The method is applied to FLUX.1 Kontext and Qwen-Image-Edit, achieving PSNR improvements of 1-3 dB while maintaining competitive editing scores. The paper also augments GEdit-Bench and ImgEdit-Bench with mask annotations and pixel-level metrics for more rigorous evaluation.

**Compliance With Llm Reviewing Policy:**

Affirmed.

**Final Justification:**

Overall, the paper tackles a relevant problem with a reasonable and well-motivated technical design, supported by ablations. The introduction of new datasets and benchmarks is also a valuable contribution, and the paper is generally clear with helpful visualizations.

The rebuttal has adequately addressed my main concerns through additional clarifications and supporting experiments. I encourage the authors to incorporate these details into the final version. Based on this, I maintain my original positive score.

**Key Questions For Authors:**

**Q1.** The current ablation does not isolate individual components. Can you provide results for the pixel-level reward alone without the MLLM reward, L+_ner and L-_er applied separately, and varying dataset sizes? This would clarify which design choices matter most.

**Q2.** How does CoCoEdit compare to non-RL editing methods on the same benchmarks? This would help determine whether the consistency gains require the RL formulation or could be achieved otherwise.

**Q3.** Since CoCoEdit-40K is a key component, what performance does supervised finetuning on the same dataset achieve? This comparison is necessary to separate the effect of data curation from the proposed RL training framework.

**Limitations:**

yes

**Strengths And Weaknesses:**

### Strengths

**S1: Relevant Problem**
The paper targets the trade-off between editing quality and content consistency, which is supported by the quantitative gap shown in Figure 1 and Table 1.

**S2: Reasonable Technical Design**
The pixel-level similarity reward combined with region-based regularizers offers a straightforward way to address spatial-agnostic reward limitations. The ablation in Figure 7 provides some validation of design choices.

**S3: Dataset and Benchmark Extension**
The CoCoEdit-40K dataset and the mask-augmented benchmarks (GEdit-Bench, ImgEdit-Bench) with pixel-level metrics could be useful resources for future work on content-consistent editing.

**S4: Generally Clear Writing**
The paper is reasonably well-organized, and the visual comparisons in Figures 2 and 6 help illustrate the method's behavior.

### Weaknesses

**W1: Incomplete Ablation Studies.**
The ablation in Figure 7 only examines reward ratios and the presence or absence of region-based regularizers. It does not isolate the pixel-level reward without the MLLM reward, nor does it separate the individual contributions of L+_ner and L-_er. The impact of dataset size and sensitivity to other hyperparameters beyond the reward ratio are also unexplored. A more thorough ablation would better clarify how much each component actually contributes.

**W2: Limited Comparison with Non-RL Methods.**
The paper primarily compares with RL-based methods such as Edit-R1 and MotionNFT. Including non-RL editing approaches would better position the work and clarify whether the consistency gains are specific to the RL paradigm or achievable through other means. Specifically, The improvements may partly stem from the curated CoCoEdit-40K dataset rather than the proposed RL framework itself. The authors should include results of supervised finetuning on the same dataset to disentangle the contribution of data quality from the training algorithm.

---

> ### Author Rebuttal · Authors · 2026-03-29
>
> We sincerely thank this reviewer for the constructive comments. Please find what below our point-to-point responses.
>
> `Q1. Ablation studies`
>
> Thanks for the nice suggestions on the ablation studies. Our detailed responses are as follows.
>
> **1. Isolation of Individual Components:**
>
> We respectfully direct this reviewer to **Appendix C.3, Tab. 4**, where we have already provided a detailed ablation study isolating the individual contributions of the pixel-level similarity reward, $L_{ner}^+$, and $L_{er}^-$. The findings in Tab. 4 are summarized below:
>
> *  **Baseline (MLLM reward only):** Achieves good editing scores but suffers from poor consistency (low PSNR).
> *  **w/ Pixel-level Reward:** Brings a noticeable but limited improvement in PSNR, showing that it can guide the model toward structural integrity.
> *  **w/ $L_{ner}^+$:** Leads to the highest PSNR with a slight drop in the overall score, indicating that penalizing the background too strictly makes the model over-conservative.
> *  **w/ $L_{er}^-$:** Effectively complements the positive term by encouraging edits in the target region. This setting achieves the optimal balance.
>
> Regarding the setting of using **only pixel-level similarity reward** (without MLLM reward or other components), we did not include this in our report because the training process will collapse under this configuration.
> Without using the MLLM reward to drive the actual editing effect, the model prioritizes visual consistency and ignores the editing effect. Empirically, we observe that the similarity reward increases rapidly during training as the model finds a "shortcut." Consequently, the model converges to a trivial identity mapping. The output of the test sample is identical to the input image, completely ignoring the editing instructions. Thus, we set the MLLM reward as a default component.
>
> **2. Impact of Dataset Size:**
>
> Unlike SFT, RL training does not require massive datasets to converge, as the model learns from reward-driven exploration rather than fitting ground-truth patterns. The quality is more important than scale. Prior works like Edit-R1 and MotionEdit also use relatively small datasets (e.g., ~27K and ~11K).
> To empirically validate this, we conducted an additional ablation using a larger dataset. As shown below, scaling up the data size yields little improvement in performance, confirming that our 40K high-quality dataset is sufficient for the RL optimization process.
>
> |Dataset Size|GEdit(Overall)|GEdit(PSNR)|ImgEdit(Overall)|ImgEdit(PSNR)|
> |:---|:---:|:---:|:---:|:---:|
> |40K(Ours)|**7.754**|**22.283**|**3.79**|19.125|
> |120K|7.723|22.204|**3.79**|**19.201**|
>
> `Q2. Comparison with Non-RL methods`
>
> We would like to clarify that, in Tabs. 1&2 of the main paper, methods such as **BAGEL, OmniGen2, and Step1X-Edit** are all non-RL models. CoCoEdit significantly outperforms them in both editing quality and content consistency.
>
> `Q3. Comparison with SFT on CoCoEdit-40K`
>
> This reviewer raises an excellent point regarding disentangling the contribution of our data curation from the RL algorithm. However, **direct SFT on CoCoEdit-40K is not suitable because of two reasons**. First, our curated triplets (Original Input, Dilated Mask, Refined Instruction) do not have GT. Furthermore, during our data filtering pipeline, we only evaluate instruction clarity, mask accuracy, and target prominence. When curating an SFT dataset, the filtering criteria would have to be different (e.g., the instruction following and visual quality of GT).
>
> To differentiate the contributions of the dataset and the RL algorithm, we extract the GT images of samples in CoCoEdit-40K from source datasets and perform SFT. To make it fair, we add a consistency loss (calculating PSNR/SSIM on the non-edited regions between the output and GT) during training.
> As shown in the table below, while the SFT baseline achieves a slight improvement in consistency metrics over the base model due to the explicitly added consistency loss, its Overall editing quality actually drops. This can be expected since our data curation pipeline is specifically tailored for RL and does not filter for the visual quality or instruction following of GT images, which are critical for effective SFT. In contrast, our CoCoEdit significantly outperforms both the base model and the SFT baseline across all metrics. This clearly demonstrates that the substantial gains in both editing quality and consistency stem primarily from our proposed RL training algorithm and reward formulation, rather than merely from the curated data itself.
>
> |Method|GEdit(Overall)|GEdit(PSNR)|ImgEdit(Overall)|ImgEdit(PSNR)|
> |:---|:---:|:---:|:---:|:---:|
> |Qwen-Image-Edit|7.560|19.488|3.70|17.635|
> |w/SFT|7.219|20.293|3.61|18.048|
> |w/CoCoEdit|**7.754**|**22.283**|**3.79**|**19.125**|
>
> We will include the above discussion and the SFT comparison in the revised manuscript.

---

> > ### Author Rebuttal · Reviewer_UsUV · 2026-04-03
> >
> > We thank the authors for the thorough response. Our concerns are satisfactorily addressed, and we maintain our positive assessment.

---

> > > ### Author Response · Authors · 2026-04-03
> > >
> > > We sincerely thank this reviewer for acknowledging that your concern is solved! We will surely incorporate your constructive comments in the revision of the manuscript.

---

### Official Review · Reviewer_VpFt · 2026-03-08

**Soundness:** 3
**Presentation:** 3
**Significance:** 3
**Originality:** 3
**Overall Recommendation:** 4
**Confidence:** 4

**Summary:**

This paper introduces CoCoEdit, a post-training framework designed to resolve the inherent conflict between editing effectiveness and content consistency in generative models. Experimental results on enhanced benchmarks demonstrate that CoCoEdit significantly improves the content preservation of state-of-the-art models like FLUX.1 Kontext and Qwen-Image-Edit.

**Compliance With Llm Reviewing Policy:**

Affirmed.

**Final Justification:**

I maintain my original positive recommendation after reviewing the rebuttal. My key concern that using PSNR/SSIM rewards favors pixel-level performance and weakens methodological novelty remains. However, the authors sufficiently clarified experimental baselines, the consistency-quality trade-off, and how over-regularization is avoided. The work is technically solid and practically useful, so I keep my positive score.

**Key Questions For Authors:**

See weakness.

**Limitations:**

Yes

**Strengths And Weaknesses:**

Strengths
1. The development of CoCoEdit-40K provides a robust, large-scale training set with precise mask annotations and refined instructions, filling a gap in region-aware image editing data.
2. The introduction of distinct positive and negative regularizers effectively addresses the common pitfall where consistency constraints lead to conservative or insufficient editing.

Weaknesses
1. The reliance on pixel-level metrics such as PSNR and SSIM to determine the final Rank in Tables 1 and 2 is theoretically questionable. Image editing is primarily a semantic and generative task where visual quality and instruction following should take precedence over exact pixel-wise alignment. Notably, the Overall editing scores indicate that CoCoEdit performs similarly to, and in some instances worse than, Edit-R1. This suggests that the pixel-level reward may act as a restrictive constraint that inadvertently degrades the global editing quality or creative capacity of the model.
2. A significant concern regarding the experimental validation arises from Table 4 in Appendix C.3. The baseline performance (without the inclusion of the pixel-level similarity reward or the positive/negative regularizers) already exceeds the SOTA results presented in the main text (Tables 1 and 2). This discrepancy suggests an uneven comparison, making it difficult to discern whether the observed improvements are a result of the proposed RL framework or simply the superiority of the underlying base model and general fine-tuning environment.
3. The use of PSNR and SSIM to measure consistency in non-edited regions may lead to over-regularization. These metrics are notoriously sensitive to minor pixel shifts or subtle lighting adjustments that are often perceptually imperceptible.

---

> ### Author Rebuttal · Authors · 2026-03-29
>
> We sincerely thank this reviewer for the constructive comments. Please find what below our point-to-point responses.
>
> `Q1: Pixel-level metrics and the trade-off with editing score`
>
> We agree with this reviewer that image editing is a semantic and generative task. However, for **local editing tasks**, it is expected that the editing effects are only applied to the target itself, while the contents of unedited regions are preserved without changing. While SOTA models and RL methods (like Edit-R1) achieve high semantic editing scores, they often produce unexpected modifications in non-edit regions. The issues of pixel shift and incorrect spatial structure are conspicuous in the results of existing methods, which is unexpected during the editing application. Furthermore, in applications requiring multi-turn editing, even if the pixel change is minor in one turn, the unwanted background changes will accumulate and eventually destroy the image content. Therefore, we use MLLM scores to evaluate the semantic editing quality, and use PSNR/SSIM strictly on the **non-edit regions** to evaluate content consistency.
>
> Indeed, the pixel-level reward will constrain the editing quality. This is actually the main challenge that we addressed in this work. With a slight drop in overall editing score compared to Edit-R1, our CoCoEdit achieves significantly higher content consistency. Note that Edit-R1 aggressively modifies the image content (low similarity score), since MLLMs often fail to evaluate the consistency of image content. CoCoEdit trades a very marginal drop in editing score for a massive gain in background preservation, which we argue is a much more practical and desirable choice, especially for local editing tasks.
>
> `Q2: Results in Tab. 4 and Tab. 1 & 2`
>
> Sorry for the confusion. To clarify, the first row of Tab. 4 (the "baseline") shows the results of Qwen-Image-Edit trained on our proposed CoCoEdit-40K dataset using only the MLLM reward (the same reward used in Edit-R1). This ablation is specifically designed to demonstrate that simply constructing a dataset to apply RL training is **not enough** to solve the consistency problem. As shown in Tab. 4, when trained only with the MLLM reward, the model suffers a severe drop in content consistency (e.g., PSNR shows 18.236 dB on GEdit-Bench, which is lower than all baseline methods in Tab. 1).
> By introducing our proposed **pixel-level similarity reward and region-based regularizers** (full CoCoEdit) to utilize our CoCoEdit-40K dataset, the PSNR jumps significantly to 22.283 dB (ranking the 3rd among all 10 methods in Tab. 1) and 19.125 dB on ImgEdit (ranking the 4th in Tab. 2). This shows that the observed improvements in consistency mainly come from our proposed RL framework, rather than the dataset or base model. We will make the description of this baseline more explicit and clearer in the revised manuscript.
>
> `Q3: Potential over-regularization by using PSNR and SSIM`
>
> Directly using PSNR or SSIM as a pixel-wise loss function typically leads to over-smoothing and over-regularization, as it penalizes imperceptible shifts. Our CoCoEdit solves this issue by using PSNR and SSIM only in reward calculation. We do not use PSNR/SSIM as direct optimization targets (losses). Instead, we compute them to form a single **scalar reward** $r_{sim}$ for each sample. This scalar is used to weight the overall loss of the trajectory in RL training (Eqs. 2 & 6). This mechanism relaxes the strict pixel-matching constraint; it merely increases the probability of sampling trajectories that preserve the background better, without forcing the network to memorize exact pixel values.
>
> Furthermore, for our region-based positive regularizer $L_{ner}^+$, we explicitly introduce a tolerance threshold $\tau^+$ (Eq. 4 and Appendix B.3.1). The penalty is activated only when the background drift exceeds this threshold. Minor pixel shifts or subtle lighting adjustments are ignored, preserving the model's generative diversity and preventing over-regularization.

---

> > ### Author Rebuttal · Reviewer_VpFt · 2026-04-02
> >
> > Thank you for your detailed response to my concerns. I hold a consistent view with that of Reviewer 2MyZ: using PSNR and SSIM as training rewards will naturally yield advantages on these pixel-level metrics over mainstream methods, which diminishes the highlight of performance improvements brought by your methodological contributions. Even so, I will keep my original positive score for this work.

---

> > > ### Author Response · Authors · 2026-04-02
> > >
> > > We sincerely thank this reviewer for the recognition of our work and for maintaining the positive score!
> > >
> > > Regarding your remaining concern about using PSNR and SSIM as both training rewards and evaluation metrics, we would like to provide further clarification:
> > >
> > > *   **Motivation for the Reward Design:** We incorporated PSNR and SSIM into the RL reward function because they provide direct and effective pixel-level similarity feedback. This explicitly guides the model to produce results with higher content consistency through RL training.
> > > *   **Content Consistency Improvement:** The significant improvements in PSNR and SSIM over the baselines are fundamentally the result of the enhanced consistency achieved by our CoCoEdit training framework. The high PSNR and SSIM scores reflect actual background preservation, rather than merely overfitting to the reward metrics.
> > > *   **Generalization to Unseen Metrics:** To further substantiate this, as shared in our response to Reviewer 2MyZ, we evaluate our method and baselines using **LPIPS** and **DINO structural distance**. Importantly, **neither of them is used as the training reward**. CoCoEdit still demonstrates a significant advantage on these metrics, confirming that our method genuinely improves the content consistency of the edited results.
> > > *   **Balanced Editing Quality and Consistency:** Furthermore, although we incorporated PSNR and SSIM as rewards, our **MLLM Overall Score** remains competitive. In fact, in most cases, it even surpasses Edit-R1. This clearly indicates that we are not simply overfitting to PSNR and SSIM to artificially inflate our quantitative results on consistency metrics. Instead, our CoCoEdit establishes a well-balanced training framework that effectively harmonizes editing quality with content consistency.
> > >
> > > |Model|GEdit (Overall)|GEdit (PSNR)|GEdit (SSIM)|GEdit (LPIPS)|GEdit (DINO)|ImgEdit (Overall)|ImgEdit (PSNR)|ImgEdit (SSIM)|ImgEdit (LPIPS)|ImgEdit (DINO)|
> > > |:---|:---:|:---:|:---:|:---:|:---:|:---:|:---:|:---:|:---:|:---:|
> > > |FLUX.1 Kontext|6.286|24.168|0.825|0.1502|0.8714|3.43|19.591|0.592|0.3031|0.7350|
> > > |w/ Edit-R1|**7.113**|19.013|0.716|0.2141|0.8040|3.51|15.722|0.506|0.3558|0.6704|
> > > |w/ CoCoEdit (Ours)|6.939|**25.331**|**0.874**|**0.1393**|**0.8819**|**3.52**|**20.747**|**0.635**|**0.2974**|**0.7431**|
> > > ||||||||
> > > |Qwen-Image-Edit|7.560|19.488|0.662|0.1849|0.8305|3.70|17.635|0.526|0.3587|0.6723|
> > > |w/ Edit-R1|7.746|18.441|0.639|0.2136|0.8039|3.72|16.130|0.482|0.3941|0.6338|
> > > |w/ MotionNFT|7.711|18.709|0.642|0.2014|0.8128|3.71|17.072|0.514|0.3693|0.6595|
> > > |w/ CoCoEdit (Ours)|**7.754**|**22.283**|**0.774**|**0.1618**|**0.8522**|**3.79**|**19.125**|**0.555**|**0.3313**|**0.6943**|
> > >
> > > As shown above, the comprehensive performance of CoCoEdit across both the MLLM Overall Score and all consistency metrics (PSNR, SSIM, LPIPS, and DINO) fully demonstrates its significant advantage in content-consistent editing. We will include these additional discussions and the LPIPS/DINO results in the revised manuscript.
> > >
> > > Thanks again for the valuable time and constructive feedback!

---

### Official Review · Reviewer_jMaD · 2026-03-08

**Soundness:** 3
**Presentation:** 3
**Significance:** 3
**Originality:** 3
**Overall Recommendation:** 4
**Confidence:** 4

**Summary:**

This paper addresses the critical challenge of maintaining content consistency in instruction-based image editing, where existing models often introduce unintended changes to non-target regions. While the problem of "over-editing" or "spatial leakage" is known, the authors argue that current RL-based post-training methods exacerbate this by prioritizing editing strength over preservation. To solve this, the authors propose CoCoEdit, a framework featuring a dual-reward system (MLLM-based and pixel-level similarity) and region-based regularizers ($L_{ner}^{+}$ and $L_{er}^{-}$) to decouple constraints for edited and non-edited areas. They also contribute CoCoEdit-40K, a high-quality dataset of 39.7K triplets with refined instructions and dilated masks. Extensive experiments on GEdit-Bench and ImgEdit-Bench demonstrate that CoCoEdit significantly improves content consistency (measured by PSNR/SSIM) while maintaining or improving editing quality.

**Compliance With Llm Reviewing Policy:**

Affirmed.

**Final Justification:**

I thank the authors for their excellent and thorough rebuttal. I am maintaining my positive score.

**Key Questions For Authors:**

1. In Equation 5, the authors mention $\tau^{-}$ is an adaptive threshold. Could the authors clarify the exact heuristic or calculation used to determine this threshold for different edit types (e.g., how does it differ between "Add" and "Change Action")?
2. Given the limitation regarding over-smoothing in the edit region, have the authors considered a perception-based loss (like LPIPS) within the regularizer to maintain local texture while allowing for attribute changes?
3. Could the authors provide a comparison of the training time/VRAM usage of CoCoEdit versus the standard Edit-R1 baseline? Does the region-based masking add significant latency to the policy optimization step?
4. Since the masks are only used in training, does the model develop an internal spatial bias? How does it perform on images with extremely cluttered backgrounds where "non-edit" regions are visually similar to "edit" targets?

**Limitations:**

Yes. The authors provide a dedicated section discussing failure cases related to physical plausibility (scale/perspective) and the trade-off between attribute editing and texture preservation. They also acknowledge the impact of MLLM hallucinations on dataset quality.

**Strengths And Weaknesses:**

**Strengths:**
- The method effectively adapts the DiffusionNFT/Flow-Matching objective to incorporate spatial awareness via region-based regularizers, providing a more granular control mechanism than global scalar rewards.
- By complementing MLLM rewards (which handle high-level semantics) with pixel-level similarity rewards (PSNR/SSIM in non-edit regions), the model captures fine-grained inconsistencies that MLLMs typically overlook.
- The data construction pipeline is rigorous, utilizing Qwen2.5-VL and SAM 2 for mask annotation, followed by a strict filtering process (threshold > 9.4) to ensure high-quality training triplets.
- The paper provides a comprehensive evaluation using both automated metrics and a human user study with 30 volunteers, showing a clear win in human preference and a substantial boost in PSNR (e.g., +2.8dB for Qwen-Image-Edit).
- Although trained on local editing tasks, the model shows improved performance on global editing tasks like "Style" and "Tone" which require structural preservation.

**Weaknesses:**
- While masks are not required at inference time, the training phase is heavily dependent on the accuracy of the masks generated by SAM 2 and the refinement by MLLMs. Errors in initial mask boundaries could propagate into the regularizers.
- The authors acknowledge a trade-off where strong consistency constraints can lead to over-smoothing or loss of texture within the edited region itself, particularly during strong color alterations.
- The core RL strategy builds upon existing paradigms like DiffusionNFT and GRPO. The primary innovation is the spatial decoupling and the specific formulation of the $L_{ner}^{+}$ and $L_{er}^{-}$ terms.
- While the authors compare against Step1X-Edit and Edit-R1, comparisons against some state-of-the-art in-context or mask-guided (inpainting) editors [1, 2] (even if CoCoEdit is training-based) would further contextualize the performance gap.
- The paper does not cite or discuss FireEdit [3]. Since both works aim for fine-grained editing, the authors should clarify how their RL-based region regularization differs from FireEdit's VLM-driven spatial grounding approach.

[1] black-forest-labs/FLUX.1-Fill-dev

[2] Zhang, Z., Xie, J., Lu, Y., Yang, Z., & Yang, Y. (2025). Enabling instructional image editing with in-context generation in large scale diffusion transformer. In The Thirty-ninth Annual Conference on Neural Information Processing Systems.

[3] Zhou, J., Li, J., Xu, Z., Li, H., Cheng, Y., Hong, F. T., ... & Liang, X. (2025). Fireedit: Fine-grained instruction-based image editing via region-aware vision language model. In Proceedings of the Computer Vision and Pattern Recognition Conference (pp. 13093-13103).

---

> ### Author Rebuttal · Authors · 2026-03-28
>
> We sincerely thank this reviewer for the constructive comments. Please find what below our point-to-point responses.
>
> `Q1. Mask accuracy`
>
> To mitigate the issue of inaccurate masks from SAM 2 and MLLMs, we dilate the masks during training (as mentioned in Sec. 3.1). Pixels in the mask boundary are excluded from the penalty and reward. Therefore, the training process is highly robust to minor mask imperfections. We will add more implementation details in the revision.
>
> `Q2. Consistency trade-off`
>
> The reason for this trade-off is that the pixel-level consistency reward provides a very explicit and dense supervision signal. The model tends to "over-optimize" this pixel-level reward at the expense of the MLLM reward. Therefore, balancing the weights of these two rewards is important. Through empirical tuning (as shown in Sec. 4.4), we find a balanced weight configuration that largely avoids over-smoothing while maintaining strong background consistency.
>
> `Q3. Core strategy`
>
> The comments are accurate. While we build upon DiffusionNFT, our primary contributions, including spatial decoupling, the reward and the regularization terms, address a critical bottleneck of image editing: the conflict between editing effect and content consistency.
>
> `Q4. More comparisons`
>
> As suggested, we have evaluated FLUX.1-Fill-dev and ICEdit. As shown below, FLUX.1-Fill-dev achieves good consistency scores due to the mask guidance. However, it has low overall editing scores (e.g., 3.258 on GEdit). ICEdit has better editing effects, but its consistency is poor. We will add the results in revision.
>
> |Method|GEdit(Overall)|GEdit(PSNR)|GEdit(SSIM)|ImgEdit(Overall)|ImgEdit(PSNR)|ImgEdit(SSIM)|
> |:---|:---:|:---:|:---:|:---:|:---:|:---:|
> |FLUX.1-Fill-dev|3.258|**26.793**|0.786|2.04|**26.463**|**0.701**|
> |ICEdit|5.898|21.698|0.730|2.88|21.561|0.614|
> |FLUX.1Kontext w/ CoCoEdit|6.939|25.331|**0.874**|3.52|20.747|0.635|
> |Qwen-Image-Edit w/ CoCoEdit|**7.754**|22.283|0.774|**3.79**|19.125|0.555|
>
> `Q5: Discussion of FireEdit`
>
> Although FireEdit also aims for fine-grained editing, it requires explicit spatial grounding at **inference time**. It relies on an LLM to encode bounding boxes provided by a region proposer, which are then fed into the diffusion model. Instead, CoCoEdit uses masks as spatial guidance **during training** to enhance the model's intrinsic consistency capabilities. In inference, CoCoEdit is completely **mask-free**. We will cite FireEdit and discuss it in revision.
>
> `Q6: Calculation of $\tau^-$`
>
> As detailed in Appendix B.3.2 (Line 836), $\tau^-$ is not determined by a specific task category. Instead, it is dynamically defined based on the latent distance between samples $x_0$ and input images $c_I$: $\tau^- = \|P_{er}(x_0) - P_{er}(c_I)\|_2$. This makes the threshold adaptive to the magnitude of the change to be attempted by the model for that specific image, rather than relying on a constant value.
>
> `Q7: LPIPS loss`
>
> Incorporating LPIPS may help preserve local textures, while it reduces computational efficiency. The diffusion loss operates in the latent space, and calculating LPIPS needs to decode the latents back into images. This will drastically increase memory usage and significantly slow down training. Given that the over-smoothing issue is rare due to our balanced reward weights and it does not significantly impact overall performance, we choose the current latent-based formulation as a practical trade-off. We will consider integrating LPIPS in future work.
>
> `Q8: Training time and VRAM usage`
>
> Because CoCoEdit does not introduce any additional models into the RL training (masks are pre-computed and regularizers are handled via lightweight tensor operations in latent space), the **VRAM usage is identical** to standard Edit-R1 baseline (~70GB).
> Regarding training time, region-based masking and reward calculation add only a marginal latency (Edit-R1 costs ~10 min for one step and CoCoEdit costs ~12 min). We will add these comparisons in the revision.
>
> `Q9: Spatial bias and performance on cluttered scenes`
>
> To prevent the model from developing a blind internal spatial bias (i.e., relying on spatial shortcuts rather than semantic understanding), we apply **instruction refinement** during data construction (Sec. 3.1). We enrich instructions with detailed descriptions and localization cues about the editing target. This forces the model to strongly ground its edits in the semantic meaning of the prompt, allowing it to distinguish between edit and non-edit regions effectively based on text-image alignment.
>
> Regarding results on extremely cluttered backgrounds, we admit that this is a challenging issue. In such extreme cases, the model may fail to locate the target. This is a common limitation shared by current commercial models as well. Improving fine-grained semantic grounding in highly complex scenes remains an open challenge to be addressed by researchers.

---

> > ### Author Rebuttal · Reviewer_jMaD · 2026-04-03
> >
> > I thank the authors for their excellent and thorough rebuttal. They have fully addressed all of my concerns.
> >
> > Specifically, I appreciate the extra effort the authors took to run the FLUX.1-Fill-dev and ICEdit baselines, they provide great context and clearly highlight the balanced strength of CoCoEdit. Their practical justification for omitting the LPIPS loss due to VRAM and latency constraints is perfectly reasonable. Furthermore, the clarifications regarding the adaptive threshold, the comparison to FireEdit, and the exact training overhead metrics are very helpful.
> >
> > I am maintaining my positive score.

---

> > > ### Author Response · Authors · 2026-04-03
> > >
> > > We sincerely thank this reviewer for acknowledging that the concerns were addressed and maintaining the positive score!

---

### Official Review · Reviewer_2MyZ · 2026-03-13

**Soundness:** 2
**Presentation:** 2
**Significance:** 2
**Originality:** 2
**Overall Recommendation:** 4
**Confidence:** 4

**Summary:**

The paper combines Diffusion NFT with mask-based regularization to balance editing capability and source consistency. The first contribution of the paper is augmenting the editing dataset which includes a mask indicating the edited region, and the second contribution of the paper is to propose applying region-dependent regularization to the NFT loss. Specifically, CoCoEdit regularizes non-edit regions to be closer to the source image for the positive group, while regularizes edited regions to be far away from the source image for the negative group. For the measure of distance from the source image, the paper utilizes PSNR and SSIM.

The experimental results demonstrate the effectiveness of the CoCoEdit in both editing fidelity and source consistency.

**Compliance With Llm Reviewing Policy:**

Affirmed.

**Final Justification:**

Because the major concerns have been addressed after rebuttal, the reviewer increases the score from 3 to 4.

**Key Questions For Authors:**

- How accurate is the down-sampled mask (which is usually downsampled by x8 or x16) in detecting the edited region?  Isn’t this causing the problem of “inaccurate judgement of MLLM for the local editing”?

- Could authors report all metrics reported in PIEBench, including background LPIPS and DINO structural distance?

- Is it fair to compare the CoCoEdit, which explicitly uses PSNR and SSIM as training reward, with other baselines that are not optimized for those metrics, when the evaluation is also based on PSNR and SSIM? If the authors believe that the results are nevertheless non-trivial, the reviewer would appreciate further clarification or justification.

- Could authors provide an ablation study with pure diffusion NFT?

- Could authors clarify the major difference of the CoCoEdit-40k with other editing datasets?

**Limitations:**

yes

**Strengths And Weaknesses:**

**Strength**
- The paper presents a clear and well-motivated problem setting.
- The proposed method achieves comparable improvements in editing fidelity with other post-training methods, while outperforms them in terms of source consistency.
- The experiments are done with two major editing models and two benchmarks, which makes the results significant.


**Weakness*

- The paper claims that MLLM-based rewards are insufficient for capturing detailed differences among edited samples. However, the method relies on masks that are downsampled to the latent resolution to distinguish edited region and non-edit region. The accuracy and reliability of these masks are not analyzed, despite their importance to the proposed method.

- The pixel-similarity reward is a combination of PSNR and SSIM, which is not a novel design. Moreover, it is somewhat problematic to use only these metrics for evaluation, as the model is trained to improve PSNR and SSIM of non-edit regions. It is expected to outperform other methods which are not trained to maximize those metrics. The paper also does not report other metrics for the source consistency used in PIEBench, such as background LPIPS and DINO structural distance.

- Missing ablation study with pure diffusion NFE without regularizations (i.e. lambda_ner = lambda_er =0)

- The contribution of the dataset should be clarified. The main paper does not sufficiently discuss how CoCoEdit-40k differs from existing datasets in terms of pipeline, scale, or distinctive features.

---

> ### Author Rebuttal · Authors · 2026-03-28
>
> We sincerely thank this reviewer for the constructive comments. Please find what below our point-to-point responses.
>
> `Q1. Mask accuracy`
>
> The down-sampled masks (e.g., 64x64) are only applied to the latent features in our region-based regularizers. For diffusion models, even at the 64x64 resolution, the latent space can still preserve the spatial layout and semantic structure of the image, and the mask could decouple the edit and non-edit regions to update gradients.
> When calculating pixel-similarity reward, we do it at the original image resolution. We use **full-resolution masks** to compute masked PSNR and SSIM, ensuring highly accurate pixel-level feedback.
>
> Note that the MLLM reward does not use any masks. The MLLM evaluates the whole edited image against the instructions to judge the editing effect. Therefore, current MLLMs are inherently limited in distinguishing fine-grained background details, which is the motivation for us to introduce the pixel-level reward. We will make these points clearer in the revision.
>
> `Q2. The pixel-similarity reward is not novel`
>
> We'd like to clarify that our novelty does not lie in the pixel-similarity reward design. Actually, simply using PSNR/SSIM as rewards does not solve the problem, as shown in Fig. 7. Our contribution lies in the design of the region-regularized RL paradigm, which integrates pixel-level rewards, MLLM rewards, and region-based regularizers to **significantly improve the consistency metrics without compromising the editing effect**. In contrast, previous RL-based editing methods largely sacrifice consistency, wrongly modifying many background regions.
>
> `Q3. LPIPS and DINO metrics`
>
> Thanks for the suggestion! We have evaluated our models and the baselines using Background LPIPS and DINO structural distance. The results on FLUX.1 Kontext and Qwen-image-edit backbones are shown below:
>
> |Model|GEdit(LPIPS)↓|GEdit(DINO)↑|ImgEdit(LPIPS)↓|ImgEdit(DINO)↑|
> |:---|:---:|:---:|:---:|:---:|
> |FLUX.1Kontext|0.1502|0.8714|0.3031|0.7350|
> |w/Edit-R1|0.2141|0.8040|0.3558|0.6704|
> |w/CoCoEdit|**0.1393**|**0.8819**|**0.2974**|**0.7431**|
> ||||||
> |Qwen-Image-Edit|0.1849|0.8305|0.3587|0.6723|
> |w/Edit-R1|0.2136|0.8039|0.3941|0.6338|
> |w/MotionNFT|0.2014|0.8128|0.3693|0.6595|
> |w/CoCoEdit|**0.1618**|**0.8522**|**0.3313**|**0.6943**|
>
> As shown, CoCoEdit consistently outperforms the baselines and other RL-based methods on LPIPS and DINO, further validating that our method preserves the semantic and perceptual integrity of non-edit regions. We will include these results in the revision.
>
> `Q4. Comparison fairness`
>
> We respectfully argue that our evaluation protocol is fair. Our primary objective is to solve the background inconsistency issue in current editing models. Since PSNR and SSIM are the most standard metrics for measuring pixel-level similarity, it is reasonable to evaluate all models on them. Note that our method achieves the goal of content consistency without sacrificing the editing effect, demonstrating the key advantage of CoCoEdit. In addition, the human subjective evaluation in Tab. 1&2 further confirms that users strongly prefer the outputs of CoCoEdit due to its superior background preservation, which is an independent metric not directly optimized during training.
>
> `Q5. NFT without regularization`
>
> The setting "w/ Edit-R1" in Tab. 1&2 is exactly the implementation of pure DiffusionNFT for image editing. As mentioned in Line 110 of the main paper, Edit-R1 exploits DiffusionNFT with MLLM feedback. Furthermore, in our ablation study in Tab. 4 of Appendix C.3, the second row represents results without region-based regularization. The performance drop shows the necessity of our proposed regularizers over pure DiffusionNFT.
>
> `Q6. Difference between CoCoEdit-40K and other editing datasets`
>
> We apologize for not making this clear enough. CoCoEdit-40K differs from existing supervised editing datasets in the following aspects, specifically tailored for RL.
>
> **Unique Filtering Protocol** (Line 160): Existing datasets filter data mainly based on editing quality. In contrast, we evaluate Instruction Clarity, Mask Accuracy, and Target Prominence. We focus more on the quality of the condition signals (text and mask) to ensure correct regional guidance during training.
>
> **No Need for GT**: CoCoEdit-40K consists of the input image, instruction, and mask. Since RL training learns from the generated samples via reward feedback, it eliminates the costly need for filtering high-quality edited images as GT.
>
> **Instruction Refinement for Regional Awareness** (Fig. 3): We use MLLM to supplement simple instructions by injecting spatial positioning and object attributes based on the mask. This refinement explicitly helps the model learn the semantic mapping between instruction, edit region, and non-edit region, which is crucial for content-consistent editing.
>
> We will add these discussions in Section 3.1 of the revised manuscript.

---

> > ### Author Rebuttal · Reviewer_2MyZ · 2026-04-03
> >
> > Thanks to the authors for their response.
> >
> > Regarding the fairness of the comparison, the reviewer’s main concern was that the evaluation relied on PSNR/SSIM, which were also used as rewards. Accordingly, the reviewer requested additional evaluations using LPIPS and DINO distance.
> >
> > The major concerns have now been properly addressed, and the reviewer will reflect this in the score.

---

> > > ### Author Response · Authors · 2026-04-03
> > >
> > > We sincerely thank this reviewer for acknowledging that the major concerns were addressed, and we truly appreciate that this reviewer would reconsider the score!

---

### Decision · Program_Chairs · 2026-04-30

**Decision:**

Accept (regular)

**Comment:**

This paper studies a region-regularized reinforcement learning framework for content-consistent instruction-based image editing. The reviewers found the problem well motivated, the method technically clear, and the empirical results convincing. In particular, they viewed the combination of pixel-level similarity reward, region-based regularization, curated training data, and mask-augmented evaluation as a meaningful contribution to content-consistent image editing.

In the first round, the main questions concerned the reliability of training masks, the reliance on PSNR and SSIM, the strength of the ablations, and the extent to which the gains come from data curation rather than the RL formulation itself. After reading the rebuttal and discussion carefully, I am satisfied that these concerns were addressed adequately. The authors provided useful clarification on the design choices, additional comparisons, and the practical trade-offs in training. Several reviewers explicitly noted that their concerns had been resolved and maintained positive recommendations.

I therefore recommend acceptance. The paper is technically solid and addresses a relevant problem in image editing systems.